# Quantifying the interfacial triboelectricity in inorganic-organic composite mechanoluminescent materials

Xin Pan [1,2], Yixi Zhuang [2,3] ✉, Wei He[2], Cunjian Lin[4], Lefu Mei[1], Changjian Chen[2], Hao Xue [2], Zhigang Sun[5], Chunfeng Wang [6], Dengfeng Peng [7], Yanqing Zheng [5], Caofeng Pan[8], Lixin Wang [9] ✉ & Rong-Jun Xie [2,3,10] ✉

Mechanoluminescence (ML) sensing technologies open up new opportunities for intelligent sensors, self-powered displays and wearable devices. However, the emission efficiency of ML materials reported so far still fails to meet the growing application requirements due to the insufficiently understood mechano-to-photon conversion mechanism. Herein, we propose to quantify the ability of different phases to gain or lose electrons under friction (defined as triboelectric series), and reveal that the inorganic-organic interfacial triboelectricity is a key factor in determining the ML in inorganic-organic composites. A positive correlation between the difference in triboelectric series and the ML intensity is established in a series of composites, and a 20-fold increase in ML intensity is finally obtained by selecting an appropriate inorganic-organic combination. The interfacial triboelectricity-regulated ML is further demonstrated in multi-interface systems that include an inorganic phosphor-organic matrix and organic matrix-force applicator interfaces, and again confirmed by self-oxidization and reduction of emission centers under continuous mechanical stimulus. This work not only gives direct experimental evidences for the underlying mechanism of ML, but also provides guidelines for rationally designing high-efficiency ML materials.

Mechanoluminescent (ML) materials capable of converting input mechanical stimuli to photon emission have become key functional materials in mechanical sensing and energy utilization applications[1,2]. Notably, due to the features of contactless signal delivery, stress distribution visualization, self-powered emission and excellent flexibility[3,4], ML-based sensing technologies have shown great promise in structural health diagnosis[5], electronic signatures[6], temperature sensing[7,8], mechanically driven light generators[9,10], biomechanical engineering[11,12], stress recording[13–15] and flexible electronic skin[16–20].

Since the first scientific document of the ML phenomenon in 1605[21], a large number of materials possessing mechano-to-photon conversion characteristics have been discovered. Extensive studies

[1]School of Materials Sciences and Technology, China University of Geosciences Beijing, Beijing, China. [2]College of Materials, Xiamen University, Xiamen, China. [3]Fujian Key Laboratory of Surface and Interface Engineering for High Performance Materials, Xiamen University, Xiamen, China. [4]Graduate School of Advanced Science and Technology, Japan Advanced Institute of Science and Technology, Nomi, Japan. [5]School of Materials Science and Chemical Engineering, Ningbo University, Ningbo, China. [6]College of Materials Science and Engineering, Shenzhen University, Shenzhen, China. [7]College of Physics and Optoelectronic Engineering, Shenzhen University, Shenzhen, China. [8]Beijing Institute of Nanoenergy and Nanosystems, Chinese Academy of Sciences, Beijing, China. [9]Department of Vascular Surgery, Zhongshan Hospital, Fudan University, Shanghai, China. [10]State Key Laboratory of Physical Chemistry of Solid Surfaces, Xiamen, China. ✉e-mail: zhuangyixi@xmu.edu.cn; wang.lixin@zs-hospital.sh.cn; rjxie@xmu.edu.cn

have indicated that ML materials may exhibit different types of luminescent responsiveness to input mechanical stimuli, stemming from possibly different and intricate ML mechanisms. Among them, those ML materials showing surprisingly self-recoverable emissions (to generate repeatable ML without recharging with any light source), represented by the inorganic–organic composites of ZnS:Cu/Mn phosphors dispersed in a polydimethylsiloxane matrix (namely ZnS:Cu/Mn@PDMS), exhibit prominent advantages of stable ML intensity, no pre-excitation requirements, and a much lower pressure threshold[22–24]. They are thus highly expected in flexible sensing and self-powered lighting applications[25,26]. To date, much effort has been devoted to understanding the fundamental mechano-electro-photon interaction process underlying the ML phenomenon. Consequently, several mechanism models, including the triboelectric-induced electroluminescence model[27–29] and piezoelectric-induced carrier detrapping model[30,31], have been proposed. However, the exact mechanism of the self-recoverable ML observed in specific inorganic–organic composites is still controversial. Meanwhile, an effective methodology to guide the rational design of self-recoverable ML materials is unavailable, which has undoubtedly posed a huge obstacle to the development of high-performance ML materials and advanced sensing applications.

When an inorganic–organic composite material is subjected to a mechanical action such as compression or stretching, the contact-separation and lateral slip between the inorganic (ML particles) and organic phases (organic matrix) may occur at their interfaces due to the great difference in elastic modulus between the two phases (Fig. 1a). Under mechanical actions, transient triboelectricity would be generated at those interfaces, and the triboelectricity-induced electroluminescence could account for ML in the inorganic–organic composites investigated[32]. This mechanism model reasonably explains why the generation of self-recoverable ML requires neither pre-excitation of ultraviolet light nor a typical piezoelectric crystal structure (e.g., in cases of ZnS:Cu/Mn@PDMS[33], $Sr_3Al_2O_5Cl_2$:$Ln^{3+}$@PDMS[34], and $Lu_3Al_5O_{12}$:$Ce^{3+}$@PDMS[35]). Unfortunately, because such triboelectricity occurs at the microscale interfaces in the inorganic–organic composites, direct measurement of triboelectricity remains a great challenge, and solid experimental evidences for the triboelectricity-induced electroluminescence model are yet lacking. Therefore, exploring a feasible route to evaluate the triboelectricity in inorganic–organic composites and its contribution to ML is of great significance for elucidating the self-recoverable ML mechanism and developing advanced ML sensing technologies.

In this study, we proposed a method to quantify the triboelectricity between the inorganic and organic phases (triboelectric series) in ML composites based on the idea of interface amplification (Fig. 1a). Taking a series of apatite phosphors (inorganic phase) and organic elastomers (organic phase) as examples herein, we revealed that the ML intensity of the inorganic–organic composites was positively correlated with the difference in the triboelectric series between the two phases, which can be calculated by the proposed method. Importantly, a significant increase in ML intensity by ~20 times was achieved in the material system studied, which demonstrates the feasibility of improving the ML intensity by selecting two phases with a big difference in the triboelectric series. We believe that the findings in this work may greatly accelerate the development of high-performance ML materials and ML sensing applications.

## Results

### Quantifying the interfacial triboelectricity between the inorganic and organic phases in ML materials

A test system for measuring the interfacial triboelectricity between inorganics and organics was constructed based on the literature on triboelectricity nanogenerator (TENG) studies with some modifications (Fig. 1b)[36,37]. Briefly, cyclic pressing-releasing interactions

between a pair of tested materials were generated by a reciprocating linear motor. The amounts of triboelectric charge (in nC) and voltage difference (in V) during the mechanical cycles were collected by an electric meter in short-circuit and open-circuit modes, respectively. The pairs of tested materials consisted of a phosphor ceramic sheet and a pure organic film, which had the same composition as the two phases in the studied inorganic–organic composite ML materials. Specifically, the inorganic phases were selected from a series of apatite phosphors $M_{5-x}$($PO_4$)$_3$$X$:$Eu_x$ [$M$ = Ca (C), Sr (S), Ba (B); $X$ = Cl (C), Br (B), abbreviated as $MPX$)] and ZnS:Mn (reference sample, abbreviated as ZnS). The phosphors were sieved with a 200-mesh sieve to regulate the particle size, and then were sintered into compact ceramics with a diameter of ~12 mm and a thickness of ~1 mm (Supplementary Fig. 1). Both sizes of phosphor particles and ceramics were controlled to reduce the differences in contact intimacy and contact pressure of the two phases under mechanical actions for different samples. The organic phases of polydimethylsiloxane (PDMS) and silicone (SC), the most common organic matrix in ML research, were also utilized in this work. The thickness of the organic films was fixed at ~1 mm. The other measurement conditions, including the sample surface cleanliness, sample position, reciprocating frequency, reciprocating distance of the linear motor, were strictly controlled to ensure that each pair of ceramics and organic films was subject to as same mechanical strength as possible.

The principle for measuring the interfacial triboelectricity in this study was based on the mechanism of TENG[38]. As schematically depicted in Fig. 1c, a full cycle of mechanical action can be divided into four steps according to the material contact states and charge transfer states. A typical triboelectric charge curve under the short-circuit mode is given in Fig. 1d, by taking the pair of the $Ba_5$($PO_4$)$_3$Cl:Eu (BPC) ceramic and PDMS film as an example. The interfacial triboelectrification occurs when the two materials are contacted with each other, resulting in gaining electrons for the PDMS film and losing electrons for the BPC ceramic at the contact surfaces. Due to the electrostatic induction effect, a positive charge is induced at the back surface (electrode) of the PDMS film, generating a charge flow toward the electrode of ceramics (i). When the two materials are fully contacted, the transferred charge reaches the maximum (~110 nC relative to the lowest point), and there is no potential difference between the two electrodes (ii). For the releasing step, electrostatically induced charges are gradually formed on the electrodes, and the positive charges flow toward PDMS (iii). When they are completely separated with a sufficient distance, electrostatically induced charges are stably formed (iv). It is noted that the triboelectric charge curve exhibits typical characteristics of hysteresis and oscillation because the PDMS film shows the elastic deformation[39]. In the open-circuit mode, the coupling of triboelectrification and electronic induction effect generates voltage between the two electrodes during the pressing-releasing cycles, reaching a maximum value of −45.62 V in the releasing step (Fig. 1e).

The variations in the triboelectric charge and voltage of different materials were measured using the above system (Supplementary Figs. 2 and 3). The pressing-releasing mechanical stimulus was set to be as identical as possible. Based on these results, the triboelectric charge density (TECD$_{P-M}$) between the ceramic and polymer film was calculated by the following equation:

$$\text{TECD}_{P-M} = \frac{q_{P-M}}{m} \tag{1}$$

where $q_{P-M}$ is the amount of the maximum transfer triboelectric charge (in nC) and $m$ is the contact area (in cm$^2$). The calculated TECD$_{P-M}$ values of the pairs of ceramics ($MPX$ series or ZnS:Mn) and organic films (PDMS or SC) are listed in Fig. 1f, and they varies in the range of −3.61 to 36.23 nC/cm$^2$. The values of TECD$_{P-M}$ were then used to define

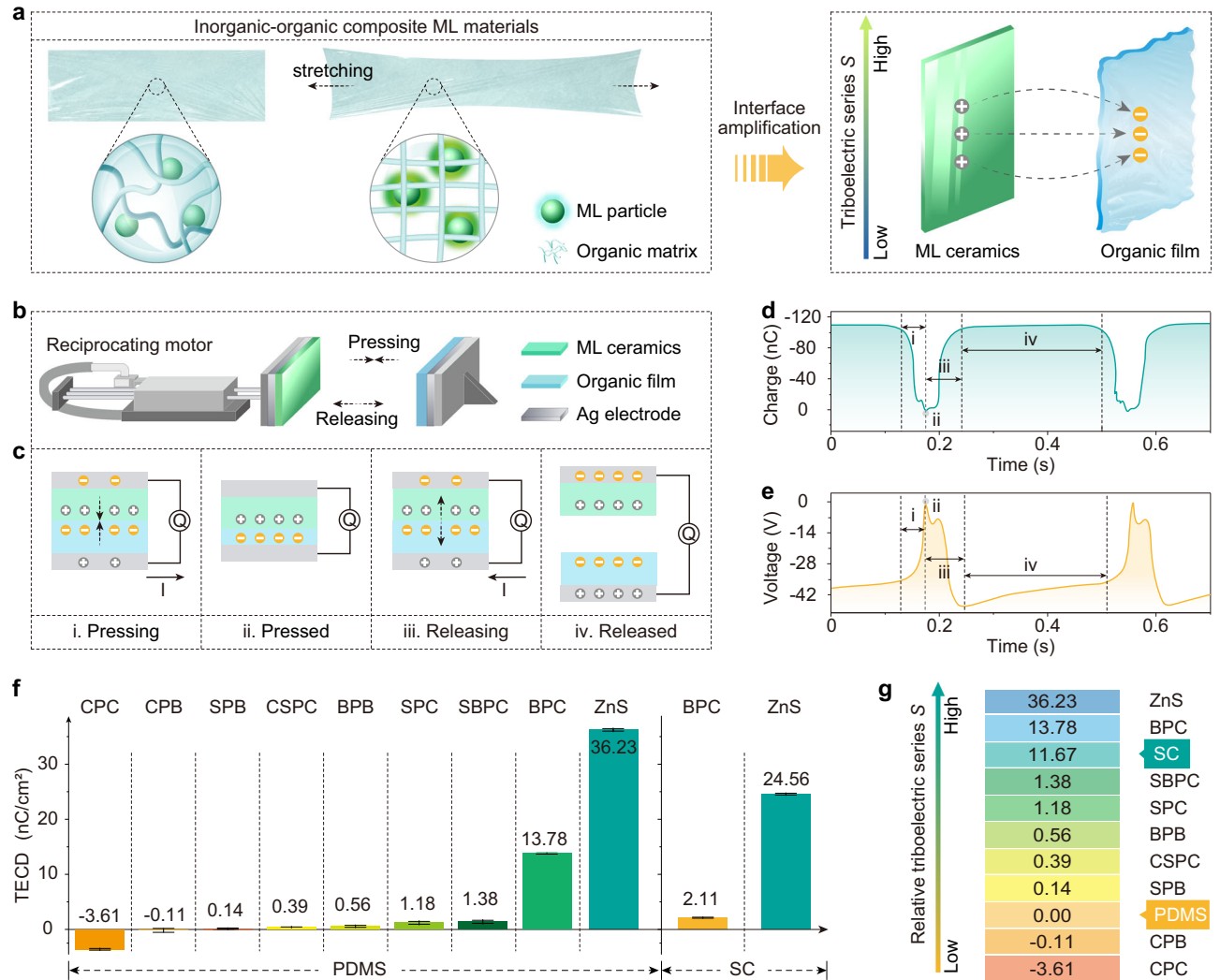

**Fig. 1 | Establishing the measurement method of relative triboelectric series for inorganic−organic ML materials. a** Structure of inorganic−organic composite ML materials, interfacial movement between inorganic and organic phases under mechanical actions, and the electron transfer between ML ceramics and organic films in the case of interface amplification. **b** Schematic diagram of the relative triboelectric series measurement system for ML. The triboelectric charge density (TECD) between inorganic ceramics and organic films was measured to evaluate the interfacial movement between inorganic phosphors and the organic matrix in composites under mechanical actions. **c** Schematic diagrams of a full cycle of mechanical actions. The generation and transition of charges due to the coupling effect of triboelectrification and electrostatic induction (in the open-circuit mode) are depicted in the figure by using the pair of the BPC ceramic and PMDS film as an example. **d** Variations in triboelectric charge (in nC) between BPC and PDMS in the pressing-releasing cycles as a typical example of TECD tests. **e** Changes in voltage (in V) in the pressing-releasing cycles. **f** TECD between inorganic phases (top) and organic phases (bottom) determined under the same mechanical action. The abbreviations of ceramics are described in the Experimental Section. Positive TECD values indicate that the inorganic phases lose electrons (i.e., the organic phases gained electrons) during friction, and negative values indicate that the inorganic phase gains electrons. $MPX$ [$M$ = C (Ca), S (Sr), B (Ba); $X$ = C (Cl), B (Br)] are abbreviations for $M_{5-x}(PO_4)_3X:Eu_x$ ceramics, ZnS is short for ZnS:Mn (reference phosphor) and SC stands for silicone. **g** Relative triboelectric series S of tested materials calculated from the results of (**f**) and Eq. (2). The relative triboelectric series $S$ of PDMS was set as the zero point. The data presented in (**f**) are shown as the mean ± 2 SD (standard deviation).

the difference in the relative triboelectricity series $\Delta S_{P-M}$ in our system[40,41]:

$$\Delta S_{P-M} = S_P - S_M = \text{TECD}_{P-M} \qquad (2)$$

Taking the relative triboelectricity series of PDMS (i.e., $S_{PDMS}$) as the zero point, the relative triboelectricity series of other ceramics and organics ($S_P$ or $S_M$) can be determined. As shown in Fig. 1g, most of the tested inorganic ceramics [except $Ca_5(PO_4)_3Br:Eu$ (CPB) and $Ca_5(PO_4)_3Cl:Eu$ (CPC)] are located at the positive side of the relative triboelectricity series, indicating that they are prone to losing electrons when compared to PDMS. Meanwhile, the SC film shows a higher $S$ value of 11.67 than many inorganic materials.

## ML properties in inorganic−organic composites

The x-ray diffraction (XRD) analysis reveals that the apatite phosphors have non-piezoelectric hexagonal crystalline structures with a space group of P21/c or P63m (Supplementary Fig. 4). After sieving, the phosphors exhibited an average particle size of ~50 mm (Supplementary Fig. 5). All the prepared inorganic−organic composites (containing the apatite phosphors) showed ML without the light pre-excitation. As presented in Fig. 2a, the ML spectra of the $MPX$@PDMS composites ($M$ = Ca/Sr/Ba, $X$ = Cl) under tensile deformation showed either only broad bands in the blue light region (due to the $4f^65d^1$–$4f^7$ transition of $Eu^{2+}$) or a combination of broad bands and sharp lines in the region of 525–700 nm (due to the $^5D_0$–$^7F_J$ transitions of $Eu^{3+}$, $J$ = 0, 1, 2, 3 and 4). The time-resolved ML intensity curves in Supplementary Fig. 6 further indicates that the ML occurs during the entire stretching-

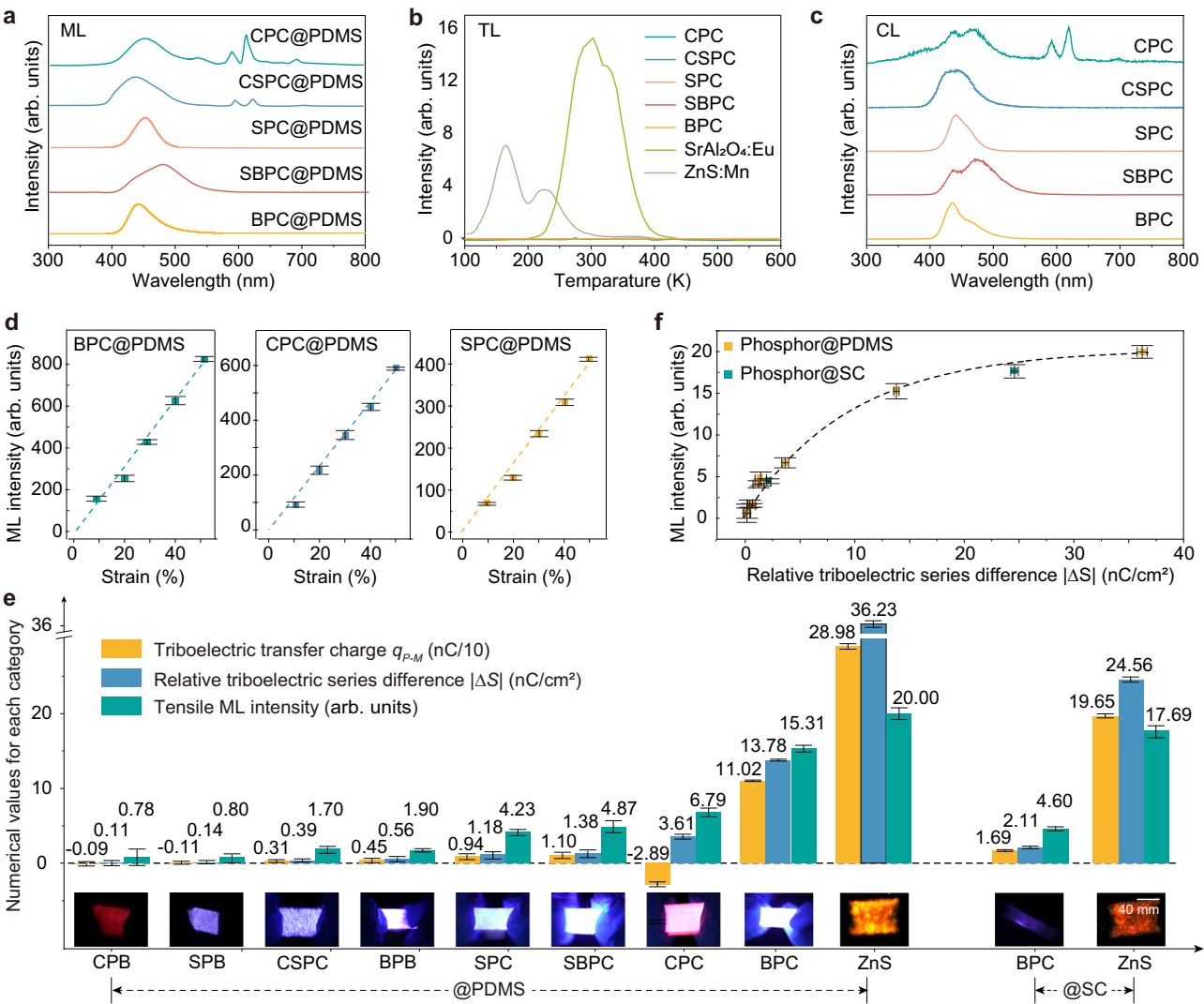

**Fig. 2 | ML properties and relative triboelectric series difference (ΔS) in inorganic–organic composite ML materials containing the apatite phosphors. a–c** ML spectra, TL glow curves and CL spectra of *MPX*@PDMS composites (*M* = Ca/Sr/Ba, *X* = Cl, luminescent center = Eu). The ML spectra were measured when 80% tensile strain was loaded on the *MPX*@PDMS films. The TL glow curves were recorded from 100 to 600 K with a fixed heating rate of 50 K/min. Prior to TL measurements, the samples were excited by 365-nm light for 20 s. The CL spectra were obtained by using scanning electron microscopy (SEM) with an electron beam of 7 kV and 60 mA. The ML and CL intensity in (**a**, **c**) is normalized to (0, 1). **d** ML

intensity of *MPX*@PDMS under different tensile strains. **e** ML intensity of the inorganic–organic composites ($I_{ML}$), triboelectric transfer charge ($q_{P-M}$), relative triboelectric series difference of the ceramic/organic pairs (ΔS), and ML image during the first stretching. *MPX* [*M* = C (Ca), S (Sr), B (Ba); *X* = C (Cl), B (Br)] are abbreviations for $M_{5-x}(PO_4)_3X$:$Eu_x$ phosphors and ZnS is short for ZnS:Mn. Photographs of the inorganic–organic composite films being stretched are inserted in the figure. **f** Correlation between ΔS and ML intensity in the tested samples. The data presented in (**d–f**) are shown as the mean ± 2 SD. 3 measurements were performed for each sample to calculate the mean value and SD.

releasing cycles and the ML intensity changes in response to the amount of deformation. There are two ML intensity peaks in a single stretching-releasing cycle. This is consistent with the TENG model that the triboelectric voltage variation has two extreme values including a peak value and a valley value (Fig. 1e).

The photoluminescence (PL) spectra of the five phosphors *MPX* (*M* = Ca/Sr/Ba, *X* = Cl) confirmed that $Eu^{3+}$ may coexist and it tends to be present in the phosphors containing Ca (Supplementary Fig. 7)[42]. This is similar with the results of the ML spectra in Fig. 2a. Meanwhile, multiple broad bands of $Eu^{2+}$ emission were observed in the apatite phosphors, especially in $Ca_{2.5}Sr_{2.5}(PO_4)_3Cl$:Eu (CSPC) and $Sr_{2.5}Ba_{2.5}(PO_4)_3Cl$:Eu (SBPC) with two alkaline earth elements. Due to the coexistence of $Eu^{3+}$ and $Eu^{2+}$ (also including multiple crystalline sites for $Eu^{2+}$), the PL spectra showed dependence on the excitation wavelength[43,44]. Furthermore, the thermoluminescence (TL) glow curves of the apatite phosphors were recorded. The TL intensity of the phosphors is negligible when compared to that of the well-known ML

phosphors $SrAl_2O_4$:Eu or ZnS:Mn (Fig. 2b). Meanwhile, pre-excitation of natural light or ultraviolet (UV) light to the apatite phosphors showed little influence on the ML intensity (Supplementary Fig. 8). These results together suggest that the ML of the *MPX*@PDMS composites should not be originated from the release of charge carriers from traps.

We further investigated the cathodoluminescence (CL) spectra of the phosphors under high-field electron bombardment (Fig. 2c and Supplementary Fig. 9)[45]. The CL spectra are basically consistent with the ML spectra (with some differences possibly due to selective excitation of different emission centers), and the CL intensity increases with increasing the kinetic energy of the applied high-field electrons. Regarding the mechanism of electroluminescence under a high field, the impact excitation model has been considered to be one of the reasonable theories to explain this process[46,47]. Our results show a positive correlation between the high-field electron energy and the CL intensity, which support the impact excitation model that more hot

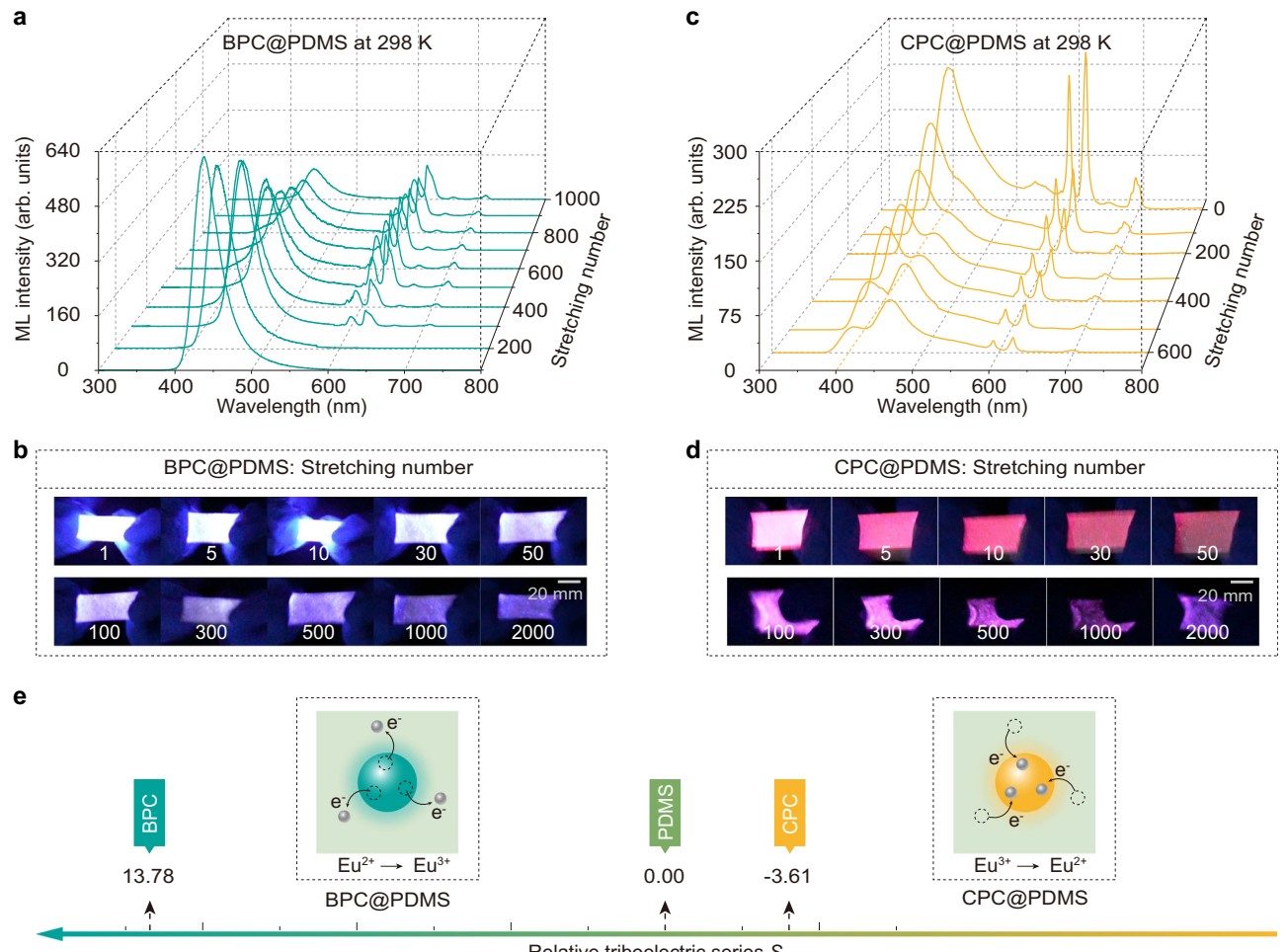

**Fig. 3 | Self-oxidization and self-reduction induced by electron transfer in ML.** **a** ML spectra and **b** photographs of the BPC@PDMS film under stretching for 2000 times. The ML was partially converted from blue (Eu²⁺) to red (Eu³⁺) under the mechanical stimulus. **c** ML spectra and **d** photographs of the CPC@PDMS film under stretching for 2000 times. Conversion of the red (Eu³⁺) to blue (Eu²⁺) emission was observed under the mechanical stimulus. The emission color of photographs with super-high brightness might slightly deviate from the original color due to the brightness saturation of the digital camera used. **e** Schematic diagrams of the electron transfer processes of ML in the BPC@PDMS and CPC@PDMS composites. Electrons transfer from BPC to PDMS by interfacial frictions because BPC possesses a higher relative triboelectric series (S) than PDMS, resulting in self-oxidization of Eu²⁺ in BPC, whereas they transfer from PDMS to CPC, leading to the self-reduction of Eu³⁺ in CPC.

electrons are generated when the impact energy increases and they further excite more luminescent centers. Also, one can find that the ML intensity of the *MPX*@PDMS composites shows a nearly linear enhancement with increasing the tensile strain (Fig. 2d).

The ML intensity of various inorganic–organic composites under the same tensile strain was measured. The integrated ML intensity of the composites ($I_{ML}$), the triboelectric transfer charge of ceramic/organic pairs ($q_{P-M}$) and their relative triboelectric series difference ($\Delta S$) are summarized in Fig. 2e. Note that the values of $q_{P-M}$ and $\Delta S$ are either positive or negative due to different charge transfer directions, and thus the absolute values of $|\Delta S|$ are used. As seen in Fig. 2f, a positive correlation can be clearly found when plotting the ML intensity ($I_{ML}$) as a function of the relative triboelectric series difference ($|\Delta S|$). It indicates that a higher ML intensity of inorganic–organic composites can be obtained by choosing a pair of materials with a larger triboelectric series difference. Impressively, the ML intensity can be enhanced by ~20 times when a suitable pair of apatite phosphors and organic matrix is chosen (e.g., from CPB@PDMS to BPC@PDMS), and BPC@PDMS shows ~75% ML intensity of the reference sample ZnS:Mn@PDMS. Meanwhile, the selection of the organic matrix is also important. In the case of BPC, the ML intensity is enhanced by ~3.3 times when PDMS with a much larger relative triboelectric series difference ($|\Delta S| = 13.78$) than that of SC ($|\Delta S| = 2.11$) is selected as the organic matrix. In addition, the intense ML in ZnS:Cu/Mn@PDMS is therefore ascribed to a much higher relative triboelectric series difference between the ZnS and PDMS phases (36.23), which therefore leads to a larger amount of triboelectricity under a mechanical stimulus (Fig. 2e, f). In contrast, the calculated triboelectric series of SrAl₂O₄ relative to PDMS is only 1.81 (Supplementary Fig. 10). Considering the large number of traps and highly efficient trap-controlled ML in SrAl₂O₄:Eu phosphors, it would be difficult to observe dominant self-recoverable ML in SrAl₂O₄:Eu@PDMS.

## Self-oxidization and reduction induced by electron transfer in ML

The self-recoverability and intensity stability of ML in the inorganic–organic composites under continuous mechanical stimulus were investigated. Taking the BPC@PDMS and CPC@PDMS films as examples, they both maintained intense ML under continuous stretching-releasing actions for 2000 times without light pre-excitation (Fig. 3a, d). A certain degree of decrease in the integrated ML intensity is generally found, owing to the structural degradation of the organic matrix under long-term stretching rather than the bad self-recoverable nature of ML[22]. With increasing the stretching number, a

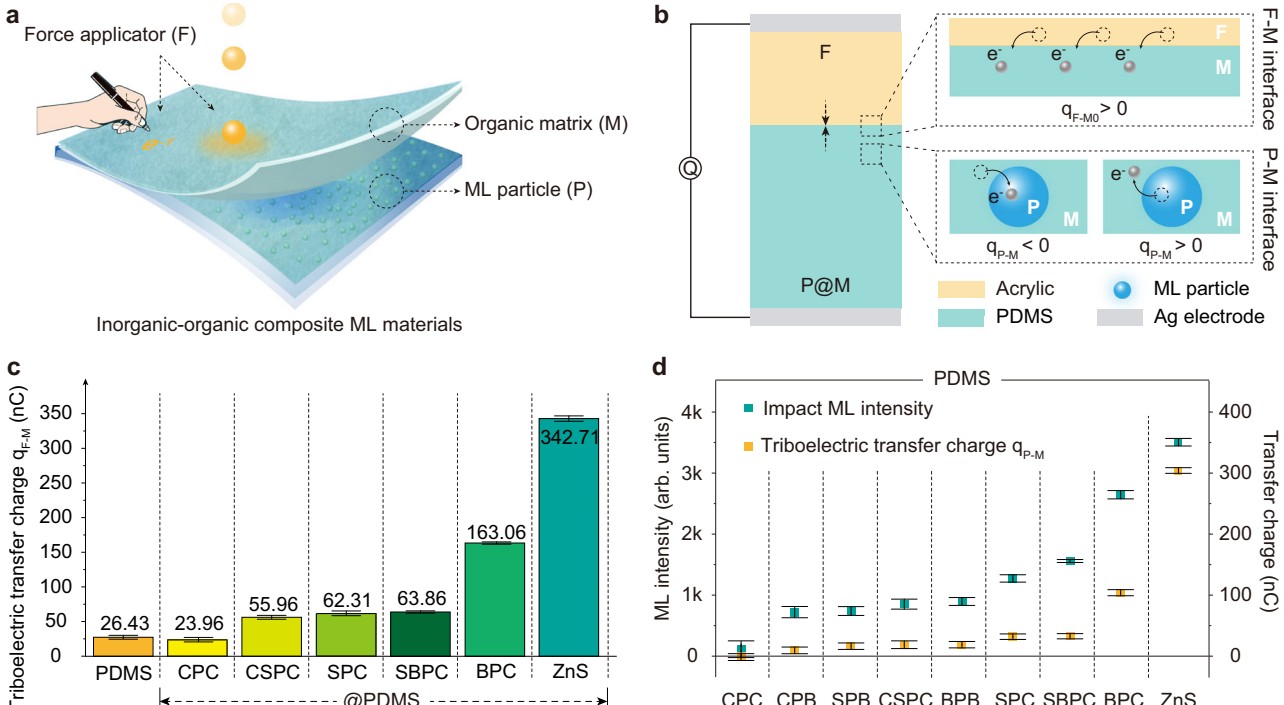

**Fig. 4 | ML and interfacial electron transfer in multi-interface systems.**
**a** Schematic diagram of a typical application of inorganic–organic composite ML materials under stress by a force applicator (F) with another type of materials. F-M represents force applicator (F)-organic matrix (M) interface and P-M represents particle (P)-organic matrix (M) interface. **b** Schematic diagram of the triboelectric charge transfer in a multi-interface system using acrylic and inorganic phosphors@PDMS composites as an example. At the F-M interfaces, the electron transfer occurs from acrylic to PDMS ($q_{F-MO} > 0$). At the P-M interfaces, different directions and different amounts of transferred charges are possible depending on the relative triboelectric series difference between the inorganic and organic phases ($q_{P-M} < 0$ or $q_{P-M} > 0$). The two interfaces together determine the total

amount of transferred charges, as shown in Eq. (3). **c** Triboelectric charge between the acrylic and different inorganic phosphors@PDMS composites. The triboelectric charge between acrylic and pure PDMS ($q_{F-MO}$) was also measured as a reference. $MPX$ [$M$ = C (Ca), S (Sr), B (Ba); $X$ = C (Cl), B (Br)] are abbreviations for the $M_{5-x}(PO_4)_3X:Eu_x$ phosphors and ZnS is short for ZnS:Mn. **d** The correlation between the triboelectric transfer (between inorganic phosphors and organic matrix, $q_{P-M}$) and impact ML intensity. The ML intensity was recorded by impacting the inorganic–organics composite films with a falling acrylic ball. The size of the acrylic ball and the height of the ball above the films were fixed in the ML measurements. The data presented in (**c**, **d**) are shown as the mean ± 2 SD. 3 measurements were performed for each sample to calculate the mean value and SD.

clear conversion from Eu²⁺ (blue) to Eu³⁺ (red) was observed in BPC@PDMS, whereas an inverse conversion from Eu³⁺ (red) to Eu²⁺ (blue) was seen in CPC@PDMS (Supplementary Movies 1 and 2). The variations in the ML spectra suggest that the luminescent center of Eu²⁺ in the BPC phosphor was self-oxidized, while that of Eu³⁺ in the CPC phosphor was self-reduced under the mechanical stimulus[48,49]. A similar phenomenon of Eu oxidation was observed in SPC@PDMS, but the degree of valence state change is smaller than that in BPC@PDMS (Supplementary Figs. 11 and 12). Furthermore, after the stretched BPC@PDMS was kept at room temperature for 30 days, the valence state change of Eu did not return to the original state, which could be seen from the comparison of their PL spectra (Supplementary Fig. 13).

To understand the self-oxidization and self-reduction processes in ML, the relative triboelectric series of the related materials (cited from Fig. 1f) are schematically depicted in Fig. 3e. As BPC possesses a higher relative triboelectric series ($S$) than PDMS, electrons will transfer from BPC to PDMS under interfacial frictions, resulting in the self-oxidization of Eu²⁺ in BPC. In contrast, electrons transfer from PDMS to CPC, leading to the self-reduction of Eu³⁺ in CPC. To the best of our knowledge, the self-oxidization and self-reduction of Eu upon continuous mechanical stimulus is firstly observed in ML materials. This phenomenon also provides solid evidence for the electron transfer process in inorganic–organic composite ML materials.

## ML and interfacial electron transfer in multi-interface systems
For most sensing applications, the inorganic–organic composite ML materials are stressed directly by another type of materials acting as a

force applicator (F) (Fig. 4a). This will introduce multiple interfaces, including force applicator-organic matrix (F-M) and inorganic phosphor-organic matrix (P-M) interfaces in the system (Fig. 4b). In such a case, the mechanical friction and electron transfer occur at both the F-M and P-M interfaces. The direction and amount of charge transfer is then determined by the relative triboelectric series of the two materials at each interface.

Herein, we used acrylic (PMMA) as the force applicator and $MPX$@PDMS composites as the ML convertors to investigate ML and electron transfer in multi-interface systems. As a reference, the triboelectric charge between the acrylic and pure PDMS ($q_{F-MO}$) was first measured, giving a positive value of 26.43 nC. Therefore, the triboelectric series of acrylic should be positive ($S_{PMMA} \sim 3.30$) with respect to PDMS, which is consistent with previous reports[50]. As shown in Fig. 4b, when PDMS is replaced by the $MPX$@PDMS composites, the total amount of transferred charges between the two electrodes will be changed due to the additional triboelectric contribution of the inorganic phosphor-organic matrix interfaces ($q_{P-M}$). The total amount of transferred charges is expressed by:

$$q_{F-M} = q_{F-M0} + \text{k} \cdot q_{P-M} \qquad (3)$$

Here, k is a factor to describe the incomplete contribution of the triboelectricity at the P-M interfaces due to the limited charge mobility in the constituent components. We assume that the k factor is a constant in different composite materials (with the same PDMS matrix and the same phosphor/PDMS ratio), the triboelectric charge between the

acrylic and *MPX*@PDMS under mechanical stimulus ($q_{F-M}$) can be estimated and the results are present in Fig. 4c and Supplementary Fig. 14. Compared with $q_{F-MO}$ of the pure PDMS, the triboelectric charge between the acrylic and *CPC*@PDMS ($q_{F-M}$) is decreased, and the triboelectric charge in other samples is increased to some degrees. These results indicate that under interfacial friction, electrons firstly transfer from the PDMS matrix to CPC particles ($q_{P-M} < 0$) or transfer from other types of inorganic phosphors (e.g., BPC or ZnS:Mn) to the PDMS matrix ($q_{P-M} > 0$), which is consistent with the relative triboelectric series in Fig. 1f. Finally, the triboelectric charge between different inorganic phosphors and the PDMS matrix ($q_{P-M}$) can be estimated according to Eq. (3), as seen in Fig. 4c. Furthermore, the number of triboelectric charges is positively correlated with the ML intensity measured by impacting the inorganic−organic composite films with a falling acrylic ball (Fig. 4d). It again verifies the contribution of the interfacial triboelectricity to the observed ML performance in multi-interface systems. Therefore, in applications that may involve multiple interfaces, a large triboelectric charge (or large relative triboelectric series difference) between the inorganic and organic phases is one of the key factors to achieve higher ML intensity in inorganic−organic composites.

Nevertheless, it should be noted that the ML in inorganic−organic composites due to the triboelectric effect can also be affected by other factors. For instance, the ML intensity can be enhanced with increasing the mechanical action frequency (i.e., deformation speed, see Supplementary Fig. 15). This is reasonable because a higher deformation speed would create more triboelectric charges within a certain duration and thus generate a higher ML intensity. Meanwhile, the ML intensity of *MPX*@PDMS can be affected by the surface smoothness of the film, and a higher smoothness leads to a lower intensity (Supplementary Fig. 16). In addition, the ML intensity is also inevitably related to the photoluminescence efficiency of the inorganic phosphors and the elastic modulus of the composites. These factors are limited within a range as possible in this work, and it is worthwhile to further investigate their effects on the ML performance in future.

## Discussion

In summary, we constructed a simple interface amplification scheme to quantify the relative triboelectric series between inorganic phosphors and organic matrix in inorganic−organic composite ML materials. Based on this scheme, the relative triboelectric series difference between the inorganic and organic phases shows a positive correlation with the ML intensity in the studied material systems. This provides solid experimental evidences to understand the ML mechanism in inorganic−organic composite ML materials and offers a clear route to explore high-intensity ML materials. Here, it is worth emphasizing that the interfacial triboelectricity makes a clear contribution to the self-recoverable ML mechanism without the need of pre-excitation (generally using the triboelectricity induced electroluminescence model), but the effects of interfacial triboelectricity on other types of ML mechanisms (such as elasticoluminescence or fractoluminescence) are not discussed in this work. We believe that for different types of ML mechanisms, the dominant factors determining the ML intensity may vary greatly.

Furthermore, we firstly report the self-oxidization or self-reduction of inorganic phosphors containing ions with variable valence states under mechanical actions, which further verifies the interfacial charge transfer associated with the triboelectric effect. Finally, triboelectric charge transfer in multi-interface systems is demonstrated to understand the ML of inorganic−organic composites in practical applications. The obtained results further indicate that the number of triboelectric charges at the interfaces of phosphors and the organic matrix is the key factor to determine the ML intensity in inorganic−organic composites. This work provides a different view to reveal the mechano-to-photon conversion mechanism as well as the

design principle of self-recoverable ML materials, which would promote the technological development of ML for advanced sensing, self-powered lighting and biomechanical engineering applications.

## Methods

### Chemicals and materials
CaCO$_3$, SrCO$_3$, BaCO$_3$ (4 N, Shanghai Ruiyu), (NH$_4$)$_2$HPO$_4$ (99.0%, Xilong Scientific), NH$_4$Br (99.0%, Dengfeng Chemicals), NH$_4$Cl (99.8%, Aladdin), and Eu$_2$O$_3$ (5 N, Baotou Rare Earth) were used as raw materials in the preparation of ML phosphors. Li$_2$CO$_3$ (97.0%, Xilong Scientific) was used as the fluxing agent for phosphors[51]. Commercialized phosphors ZnS:Mn and SrAl$_2$O$_4$:Eu were purchased from Keyan Optoelectronics and Dalian Luming. PDMS (Dowsil 184) and SC (Ecoflex00-30) were received from Dow Corning and Smooth On, respectively. Acrylic (PMMA) plates with a customized size of 40 × 20 × 2 mm$^3$ were provided by Pengbaoda Plastic Materials.

### Preparation of ML phosphors
The apatite phosphors were prepared by using a conventional solid-state reaction method. In brief, raw materials with stoichiometric compositions were mixed homogeneously and pre-sintered at 400 °C for 1 h in air to decompose the ammonium salts. After grinding into powders, the pre-sintered products were loaded in a BN crucible and further sintered in a tube furnace at 1050 °C for 5 h under a gas flow of H$_2$/N$_2$ (10%/90%). The products were naturally cooled to RT and ground into fine powders. The obtained powders were washed with acetic acid (20% in volume ratio) at 60 °C for three times and rinsed with deionized water to remove soluble impurities. The powders were collected by centrifugation, dried in air, and finally sieved with a 200-mesh sieve for further use. The stoichiometric compositions of the phosphors are $M_{5-x}(PO_4)_3 X$:Eu$_x$, abbreviated as *MPX* for simplicity. Here, *M* is selected from one or two elements of Ca (C), Sr (S), or Ba (B). *X* is selected from Cl (C) or Br (B). For example, the samples of Ca$_{5-x}$(PO$_4$)$_3$Cl:Eu$_x$ and (Sr,Ba)$_{5-x}$(PO$_4$)$_3$Cl:Eu$_x$ are denoted as CPC and SBPC, respectively. For the solid-solution compositions containing double alkaline earths (e.g., Sr + Ba), the content ratios of the alkaline earths were kept at 1:1. A series of phosphors with different doping concentrations *x* from 0.01 to 0.14 were prepared (Supplementary Fig. 17). For each combination of *M* and *X* elements in the $M_{5-x}(PO_4)_3 X$:Eu$_x$, one sample with a certain Eu concentration was chosen from the obtained phosphors based on the principle of the optimal PL intensity (Supplementary Fig. 18). This principle is adopted because it includes the effects of different host and different dopant concentration on the emission efficiency. The Eu concentration (*x* value) for CPC, CPB, SPB, CSPC, BPB, SPC, SBPC and BPC used in the study of ML-triboelectricity correlation is 0.02, 0.04, 0.04, 0.03, 0.10, 0.06, 0.08 and 0.10, respectively.

### Preparation of ML ceramics
The obtained ML phosphors were mixed with polyvinyl alcohol (PVA, in liquid) as a binder at RT and ground for ~30 min to achieve full uniformity. The ratio of PVA to phosphors was ~1 ml to 10 g. The mixtures (~ 0.3 g for one sample) were fed into a stainless-steel cylinder mold ~12 mm in diameter and pressed with a uniaxial compressive stress of ~3 MPa for 1 min. Pellet green bodies with a thickness of ~1 mm were obtained after demolding. They were sintered at 1300-1400 °C for 8 h to remove the PVA and finally formed compact ceramics.

### Preparation of inorganic−organic composite films
The obtained ML phosphors were mixed with PDMS or SC precursors to prepare inorganic−organic composite films. The weight ratios of the phosphors to the organic precursors (including the curing agents) were kept at 2:1. The ratios of the precursors to the corresponding curing agents for PDMS and SC were 10:1 and 1:1 in weight according to the product manuals. After stirring vigorously, the mixtures were

vacuumed in an oven for ~10 min to eliminate air bubbles. The bubble-free mixtures were carefully poured into an acrylic mold with a pit of $20 \times 40 \times 1$ mm$^3$ and cured at 80 °C for 5 h to obtain inorganic–organic composite films. Pure PDMS and SC films were also prepared using the same synthetic route without the use of ML phosphors. The preparation process of the composite materials is schematically depicted in Supplementary Fig. 19.

### Structural characterizations of ML phosphors and composite films

XRD patterns of the phosphors were obtained by using an X-ray diffractometer (D8 Advance, Bruker) with Cu $K_\alpha$ radiation ($\lambda = 0.15406$ nm) at 40 kV and 40 mA. The surface morphology and elemental mapping were acquired by a field emission scanning electron microscopy (SEM) instrument (SU70, Hitachi) equipped with energy dispersive X-ray (EDX) spectroscopy. The VASP package was employed to perform the structural geometry relaxation of the host matrices. Structural models were adopted from ICSD data.

### Relative triboelectricity measurements

To prepare for triboelectricity testing, the back surfaces of the materials under investigation were coated with Ag layers to serve as electrodes, achieved via sputtering for 60 s using an ultrahigh vacuum magnetron sputtering system. Post-sputtering, the samples were thoroughly cleaned with high-pressure nitrogen to remove any contaminants and then stored in a glovebox to prevent atmospheric exposure. For electrical connections during the measurements, Ag wires (99.9% purity) were used to link the electrodes with the measuring instruments.

During the actual triboelectricity measurements, we employed a unique setup where one of the samples was stationary on an optical platform while the counterpart was subjected to dynamic pressing-releasing actions. This movement was facilitated by a linear motor (H01-23 × 86/160, LinMot), programmed to a reciprocating motion of approximately 150 mm at a frequency of 2 Hz, mimicking the mechanical stress encountered in practical applications. A schematic representation of this experimental setup is detailed in Fig. 1b. The electronic signals generated from the triboelectric interactions were captured using an electrical meter (6514, Keithley) equipped with a data acquisition module (USB-6002, National Instruments), ensuring accurate measurement of the triboelectric charge and voltage. To mitigate any potential biases from initial surface charges, readings were initiated only after achieving stable signal levels through preliminary cycling.

### Optical characterizations

PL spectra were measured with a multifunctional fluorescence spectrometer (FLS980, Edinburgh Instruments). CL spectra were recorded by using a modified Mp-Micro-S instrument attached to the SEM (MonoCL4, Gatan). Temperature-dependent PL spectra were acquired with a self-built measurement system containing a UV light source of 365 nm, a fiber spectrometer (USB2000+, Ocean Optics), and a temperature controller (THMS600E, Linkam). ML spectra were recorded by using a fiber spectrometer covering the wavelength ranges of 300–1100 nm (QE Pro, Ocean Optics) under mechanical actions (e.g., stretching–releasing cycles). The integration time of the fiber spectrometers was set to be 1 s in general. For each test, the ML intensity was measured three times, and the arithmetic average of the intensity was obtained. Meanwhile, the self-recoverability of ML in inorganic–organic composites was also evaluated by using a near-field ML imaging method recently reported by our group[52] (Supplementary Fig. 20). The ML composite film was deposited directly on a complementary metal oxide semiconductor (CMOS) image sensor and excited by a free-falling ball placed at a fixed height above the film. The ML intensity generated by the falling impact was obtained by integrating the gray values or photon numbers collected by the CMOS sensor. The ML measurements were performed in a well-shielded dark room. Before the ML measurements, the samples were not deliberately exposed to any light source and they were heated to 500 K to bleach the charge carriers that might be trapped due to the charging effect of ambient light. All experiments were performed at room temperature unless otherwise specified.

### Data availability

The source data that support the findings of this study are published with the paper and Supplementary Information, and have been deposited in the Science Data Bank under accession code https://doi.org/10.57760/sciencedb.16280[53]. Any requests on data and experiments can be also available from the corresponding authors. Source data are provided with this paper.

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

## Acknowledgements

This project is financially supported by the National Key Research and Development Program [Nos. 2022YFB3503800 (R.X.) and 2022YFB3503801 (R.X.)], the National Natural Science Foundation of China [Nos. 52172156 (Y.Z.) and 51832005 (R.X.)], the Natural Science Foundation of Fujian Province of China [No. 2023J06005 (Y.Z.)] and the China Scholarship Council [No. 202306400064 (X.P)]. The authors would like to thank Prof. Wenxi Guo from Xiamen University for his valuable help in the triboelectricity characterizations of this study. The authors would like to thank Prof. Yuhua Wang, Zebin Li, Pengpeng Wang, Zhezhe Su from Lanzhou University for their supporting in the measurement of CL spectra.

## Author contributions

X.P., Y. Zhuang and R.-J.X. conceived the project. X.P., Y. Zhuang and R.-J.X. designed the experiments. Y. Zhuang, R.-J.X., L.M., D.P., Y. Zheng and C.P. supervised the research. X.P., W.H., Z.S. and C.C. were primarily responsible for the experimental system setup and data collection. X.P., C.L. and Y. Zhuang prepared the figures and wrote the main manuscript text. H.X., C.C. and C.W. worked mainly on the triboelectricity system. Z.S. and Y. Zheng contributed to ceramics preparation. X.P., C.W., Y. Zhuang and H.X. were primarily responsible for the Supplementary Information. L.W. was primarily responsible for data processing and validation. All authors contributed to data analysis, discussions and

manuscript preparation. X.P., C.L., Y. Zhuang., L.W. and R.-J.X. were responsible for revising and finalizing the manuscript.

## Competing interests

The authors declare no competing interests.
