## [Peer Review File · Nature Communications]

Quantifying the interfacial triboelectricity in inorganic-organic composite mechanoluminescent materialsReviewer #1 (Remarks to the Author):

In this paper, the authors quantified the interfacial triboelectricity generated between inorganic ML particles and organic matrices and tried to find clues about the principles of self-recoverable ML. Since the first demonstration of self-recoverable ML in ZnS:Cu microparticles embedded in PDMS in 2013, the practical aspects of ML have increased significant attention. However, the underlying mechanism remains still controversial, impeding the development of new materials. In this paper, the authors have clearly shown two findings: (1) the correlation between ML intensity and the triboelectric series and (2) the observation of charge transfer between ML particles and the matrix. I believe these findings are important for the advancement of self-recoverable ML systems and developing new ML systems with self-recoverable ML behavior is an important topic of ML research field and matches well with the scope of this journal. Therefore, I recommend the publication of this manuscript; however, several issues need to be addressed before publication.

1. Please carefully review the manuscript. The same contents are repeated in the section of [ML properties in inorganics@organics composites].

2. The authors mentioned they measured ML without the light pre-excitation. However, it is possible that fluorescent or sunlight exposure could have occurred during synthesis, sample preparation, and measurement. These light sources also contain UV and blue light components. How did the authors account for and rule out the potential influence of these sources?

3. The author mentioned the possibility that electroluminescence resulting from triboelectricity may be the cause of ML, but also claimed that ML is not originated from the trap-release process. Does this mean that electrons are excited by triboelectricity and subsequently return to the ground state, thereby generating light? Is the author considering the impact-excite model? I believe there should be further discussion on this part.

4. The phenomena of self-oxidation and reduction, as well as interfacial electron transfer in a multi-interface system, are intriguing. However, it is not clear how these phenomena relate to the paper's overall flow. I suggest that more highlighting the significance of electron transfer as experimental evidence for its occurrence between ML particle and matrix would enhance clarity and readability.

5. When does ML generation occur during the stretching-releasing of the sample? Is it during the stretching phase or at full stretch? Additionally, how many times does ML occur within a single stretching-releasing cycle? I believe providing information about these experimental phenomena could be a good reference for researchers in this field.

6. Researchers who are just beginning ML research often encounter difficulties due to the inconsistency in terminology and the many ML principles and materials. In an effort to address some of these issues, how about incorporating the term "self-recoverable" – a word that many researchers are already adopting – instead of "self-reproducible" to ensure consistency? Furthermore, I believe the paper can be a better guide for future researchers if it emphasizes difference and origins of self-reproducible (or self-recoverable) concepts compared to other ML materials.

Reviewer #2 (Remarks to the Author):

In this contribution, the authors establish a correlation between differences in triboelectric charge density and mechanoluminescence intensity in a series of inorganic/organic composite films built from apatite phosphors (doped with europium) and polymers (polydimethylsiloxane or silicone). This implies measuring the charges at the interface between the inorganic and organic parts, a task the authors achieved by the method used for preparing triboelectric nanogenerators described in the literature. Interestingly the mechanoluminescence is self-reproducible for all phosphors, so that the described systems have potentially interesting applications. For instance color change that

is demonstrated following self redox processes affecting the europium valence. Electron transfer in multi-interface systems is also demonstrated. Finally the presented data contribute to a better understanding of the mechanoluminescence generation mechanism in inorganic/organic composite materials since the presented data implying that it is most probably triboelectricity-induced electroluminescence.

Altogether, the work is well conducted, with control experiments, and the conclusions are backed by the reported data. It represents a novel and substantial contribution towards the design of highly luminescent self-reproducible mechanoluminescent materials so that I recommend publication.

Technical remarks

1. Generally speaking, experimental uncertainties should be given on reported parameters (e.g. TECD, relative values S and ΔS , tensile intensity – error bars should be associated with data points, Fig. 2d,f - qPM)
2. Captions to Figs 1f,g and 2e. Please repeat abbreviations for ceramics MPX (M = C, Ca; S, Sr; B, Ba. X = B, Br; C, Cl) so that the captions are self-contained.
3. Page 4, The text starting with "The X-ray diffraction...", left column and ending with "...mechanical stimulus (Figs 2e,2f).", right column is entirely duplicated (end of page 4, page 5, beginning of page 6).
4. Experimental part: was the Eu content of the phosphors determined experimentally too?

Reviewer #3 (Remarks to the Author):

The authors performed a systematic study on the interaction between inorganic phosphors and organic materials, in view of providing explanations for the mechanoluminescence behaviour upon application of stress. The emission mechanism of mechanoluminescence is still not very clear, especially regarding the "non-trap-based" phosphors that do not need previous excitation, before ML emission can occur. The reported experiments provide new elements for this explanation, although the exact mechanism is still not fully revealed (i.e. how the energy is transferred to the ML phosphors). Some balanced hypotheses are put forward. In my opinion, this is an important and smart work in the field and especially the proposed quantification method will have an impact in the research on mechanoluminescence.

In general the manuscript is well written, although some grammatical errors are present. At some places, the wording can be improved. For instance, in the abstract "ambiguous mechano-to-photon conversion" is not really correct, as the conversion from stress to light emission is "poorly understood" or "not yet explained", but not "ambiguous". Similarly, in the second paragraph, this is called "fantastic mechano-to-photon conversion". A more neutral tone would be better (also, I am not convinced already 'thousands of phosphors' have been shown to possess mechanoluminescence behaviour). Some terms are a bit strange, like "interface scale-up", where essentially a large interface area is intended. Finally, regarding the structure of the paper, the section "ML properties in inorganics@organics composites" contains the same text twice.

Related to the research work and the findings, the following issues should be address in more detail or substantiated.

1) I am confused about the consistency of some data sets in Figure 2. For instance, in part e, the BPC@PDMS is about 2.5 times stronger than the CPC@PDMS, whereas this is not the case in the data shown in part d. In part f, I would expect the values of the green bars in part e to be expressed as function of the values of the yellow bars in part e. However, if I look up the data points, it seems the blue bars are plotted (i.e. the triboelectric transfer charge, instead of the relative triboelectric series difference).

2) The manuscript mentions that the ML emission spectra are similar to the CL emission spectra, but both spectra (for each composition) are quite different, both in terms of the width and position of the Eu²⁺ emission band and of the presence or not of the Eu³⁺ emission. Especially the change in Eu²⁺ peak position is surprising. This point should be addressed in more detail. Also, adding (perhaps in the SI) for each phosphor composition a graph of PL, CL and ML emission would be

very helpful, in order to easily assess the differences.

3) The similarity between ML and CL spectra is used to speculate that the excitation is electrically induced. First, why is then the shape so different (i.e. in terms of the presence/absence of Eu^{3+} ; please show the CL spectra also up to 700nm), and second, what is then the possible mechanism? Can it potentially be electron impact, due to accelerated charges, similar to (high field) electroluminescence, or is it rather e-h recombination, with the energy transferred to the luminescent ions?

4) It would be very interesting to add the behaviour of $\text{SrAl}_2\text{O}_4:\text{Eu}$ to the entire discussion (perhaps with the data only in the SI to keep the story consistent). Where is this phosphor positioned in the relative triboelectric series difference? When does it transition from trap-based ML to the "self-reproducible" ML?

5) The observation of the valence state changes in the ML upon repeated stretching is a very exciting result. It would be interesting to know if it is a permanent or temporary change, when looking at the CL or PL after e.g. 1000 stretches. Regarding the statement "to the best of our knowledge, the self-oxidation and self-reduction of Eu is firstly observed...", I would guess a similar effect was already observed in Adv. Funct. Mater. 2018, 28, 1803168.

6) Regarding the self-oxidation and self-reduction, it would be interesting to add what happens to the phosphors closer to the zero value for S , but with reasonable ML intensity (e.g. for SPC, or even CPB/SPB). Is the valence state stable in that case?

A few minor issues/remarks :

7) As far as I know, the abbreviation SC for silicone is not very common. So, it would be interesting to also mention this in Figure 1, or in its caption.

8) Estimating the transferred charges is not straightforward, especially for more complex structures. What is the motivation for the validity of equation 3, considering limited charge mobility in most of the constituent components of the multi-interface system in Fig. 4b?

9) The experimental section mentions calculations of the Fermi level. However, unless I missed it, I didn't see results from those calculations.

10) The data shown in the SI is relevant. There is a good balance between the data shown in the main manuscript and in the supporting information.

Point-by-Point Response to Referees

Referee #1:

In this paper, the authors quantified the interfacial triboelectricity generated between inorganic ML particles and organic matrices and tried to find clues about the principles of self-recoverable ML. Since the first demonstration of self-recoverable ML in ZnS:Cu microparticles embedded in PDMS in 2013, the practical aspects of ML have increased significant attention. However, the underlying mechanism remains still controversial, impeding the development of new materials. In this paper, the authors have clearly shown two findings: (1) the correlation between ML intensity and the triboelectric series and (2) the observation of charge transfer between ML particles and the matrix. I believe these findings are important for the advancement of self-recoverable ML systems and developing new ML systems with self-recoverable ML behavior is an important topic of ML research field and matches well with the scope of this journal. Therefore, I recommend the publication of this manuscript; however, several issues need to be addressed before publication.

Response: First of all, we would like to thank you for your positive comments on this paper. Indeed, this paper is largely inspired by the pioneering work published in 2013 that pointed to the importance of the interfaces between inorganic particles and organic matrix for producing self-recoverable ML. We hope our paper, by providing new study methods and new experimental evidences, may further advance the ongoing research of self-recoverable ML.

We understand that there are some shortcomings in the present manuscript. We have carefully considered the issues that you mentioned and made corresponding revisions. We hope that our revisions may further meet the high standards of the journal.

Comment #1: Please carefully review the manuscript. The same contents are repeated in the section of [ML properties in inorganics@organics composites].

Response #1: Thank you very much for pointing out this problem. We apologize for our carelessness in the preparation of the manuscript. We have removed the duplicated contents and carefully reviewed the entire manuscript according to your kind suggestion.

Comment #2: The authors mentioned they measured ML without the light pre-excitation. However, it is possible that fluorescent or sunlight exposure could have occurred during synthesis, sample preparation, and measurement. These light sources also contain UV and blue light components. How did the authors account for and rule out the potential influence of these sources?

Response #2: Thank you very much for your comments on the potential influence of ambient light. Indeed, sunlight or fluorescent exposure is inevitable during the synthesis and preparation of samples. In response to this problem, prior to the ML measurements, the MPX@PDMS and ZnS@PDMS samples were heated to 500 K in this study. The heat treatment was to thermally bleach those trapped charge carriers due to the ambient light exposures and thus to minimize (if it is not possible to completely rule out) the influence of the exposures. In fact, the MPX phosphors show limited charge carrier storage capacity compared to the reference samples SrAl₂O₄:Eu or ZnS:Mn (see the TL glow curves in Fig. 2b). A comparison experiment on MPX (with and without light pre-excitation) also indicates that light exposure (either sunlight or UV

light) has little influence on the intensity of ML (Supplementary Fig. S8).

In view of the possible influence of the ML measurement conditions, we have added experimental details of the ML measurements in the Methods section. Please also see below cited from the manuscript.

“The ML measurements were performed in a well-shielded dark room. Before the ML measurements, the samples were not deliberately exposed to any light source and they were heated to 500 K to bleach the charge carriers that might be trapped due to the charging effect of ambient light.” (Methods part, Page 9)

Figure S8. Comparison of the ML intensity in BPC@PDMS with and without light exposure. (a) shows the ML intensity without light pre-excitation in the first group of 5 tests (1-5), and the ML intensity with sunlight exposure for 8 h before the second group (6-10) and the third group of tests (11-15), respectively. (b) shows the similar measurements by using UV light (10 min) as the pre-excitation source. The results indicates that pre-excitation of either sunlight or UV light has little influence on the ML intensity. The other samples containing different apatite phosphors exhibit the similar results.

Comment #3: The author mentioned the possibility that electroluminescence resulting from triboelectricity may be the cause of ML, but also claimed that ML is not originated from the trap-release process. Does this mean that electrons are excited by triboelectricity and subsequently return to the ground state, thereby generating light? Is the author considering the impact-excite model? I believe there should be further discussion on this part.

Response #3: We are grateful for your constructive comments. To be honest, this work indicates that self-recoverable ML is likely to come from interfacial triboelectricity based on a definite correlation between them, but it indeed does not provide new insights about how the triboelectricity produces ML. The mechanism of triboelectricity-induced light emission may involve complex physical effects, which unfortunately cannot be well revealed in this work. In this regard, what we can do is to refer to the classic mechanism models of electroluminescence reported in literatures, especially the impact excitation model as you mentioned (J. Appl. Phys. 1991, 70: 4505), and apply characterization methods to confirm the applicability of this model to the study of ML (e.g., Phys. Rev. B 2004, 69: 235109).

We therefore measured the CL spectra of the phosphors, which simulates the impact and excitation processes of the phosphors by high-energy electrons. The results show that the CL spectra are basically consistent with the ML spectra (with some differences due to selective excitation of different emitting centers), and the CL intensity increases with the increase of the energy of the high-field electrons (Supplementary Fig. S9). These results support the impact excitation model of the studied phosphors under triboelectricity (although we cannot rule out other mechanisms). Anyway, elucidating the mechanisms of

triboelectricity-induced light emission is beyond the reach of this work, but we are happy to add some discussion based on our experimental results. Please see the following discussion cited from the revised manuscript.

“Regarding the mechanism of electroluminescence under a high field, the impact excitation model has been considered to be one of the reasonable theories to explain this process. Our results show a positive correlation between the high-field electron energy and the CL intensity, which support the impact excitation model that more hot electrons are generated when the impact energy increases and they further excite more luminescent centers. Also, one can find that the ML intensity of the MPX@PDMS composites shows a nearly linear enhancement with increasing the tensile strain (Fig. 2d).” (Paragraph 7, Page 4)

Figure S9. Cathodoluminescence (CL) spectra of the MPX ($M = \text{Ca}, \text{Sr}, \text{Ba}; X = \text{Cl}$) phosphors by using different accelerating voltages of the electron beam. With the increase of the accelerating voltage from 3 to 7 kV, the CL intensity is increased correspondingly.

Comment #4: The phenomena of self-oxidation and reduction, as well as interfacial electron transfer in a multi-interface system, are intriguing. However, it is not clear how these phenomena relate to the paper's overall flow. I suggest that more highlighting the significance of electron transfer as experimental evidence for its occurrence between ML particle and matrix would enhance clarity and readability.

Response #4: Thank you very much for your kind suggestion. In fact, the study of self-oxidation/reduction and interfacial electron transfer in multi-interface system is to verify the contribution of the interfacial triboelectricity to ML. The obtained results well support the main argument on the correlation between triboelectricity and ML in this work, and further extend to different cases of triboelectricity-induced light emission. Therefore, we would like to highlight in these two parts the importance of electron transfer as experimental evidence for its occurrence, while without largely changing the structure of the manuscript. We hope to get your understanding. Here are relevant revisions cited from the manuscript.

“To the best of our knowledge, the self-oxidization and self-reduction of Eu upon continuous mechanical stimulus is firstly observed in ML materials. This interesting phenomenon also provides solid evidence for the electron transfer process in ML inorganics@organics composites.” (Paragraph 2, Page 6)

“Furthermore, the number of triboelectric charges is positively correlated with the ML intensity measured by impacting the inorganics@organics films with a falling acrylic ball (Fig. 4d). It again verifies the contribution of the interfacial triboelectricity to the observed ML performance in multi-interface systems.” (Paragraph 2, Page 7)

Comment #5: When does ML generation occur during the stretching-releasing of the sample? Is it during the stretching phase or at full stretch? Additionally, how many times does ML occur within a single stretching-releasing cycle? I believe providing information about these experimental phenomena could be a good reference for researchers in this field.

Response #5 Thank you again for your questions about the ML details during the stretching-releasing of the samples. Indeed, in the previous manuscript, we did not provide a fast response curve of the ML intensity variation over time, because we used a fiber spectrometer (Ocean Optics, QE Pro) with a large integrating time (usually in hundreds of milliseconds) to collect the ML spectra. To solve this problem, we modified the ML test device by using a fast response photomultiplier tube (PMT) as the photodetector (Figure S6a). With the new test device, we were able to record the time-resolved ML intensity curves during the stretching-releasing cycles. Figure S6b, S6c shows the results of BPC@PDMS as an example. Obviously, the ML generation occurs during the entire stretching-releasing process and there are two peaks in a single stretching-releasing cycle. This is well consistent with the TENG model that the triboelectric voltage variation has two extreme values including a peak value and a valley value (Figure 1e). We believe these results could be interesting and useful for further research of self-recoverable ML, and therefore we would like to add these results to the Supporting Information as Figure S6. Thank you very much for the above enlightening comments and questions.

Figure S6. Time-resolved ML measurements of samples upon mechanical stimulus. (a) Schematics of

the test device by using a fast-response PMT as the photodetector. (b) Time-resolved ML intensity curve of BPC@PDMS upon repeated stretching-releasing cycles for 60 s. This figure shows the results of the first 10 s (0-10 s) and the last 10 s (50-60 s). The repeated frequency of stretching-releasing cycles is 2.5 Hz. The maximum tensile strain is 20%. The data acquisition frequency of the static meter is set as 50 Hz. (c) The enlarged part of the first 10 cycles (0-4 s). Two intensity peaks are present in each cycle.

Comment #6: Researchers who are just beginning ML research often encounter difficulties due to the inconsistency in terminology and the many ML principles and materials. In an effort to address some of these issues, how about incorporating the term "self-recoverable" - a word that many researchers are already adopting – instead of "self-reproducible" to ensure consistency? Furthermore, I believe the paper can be a better guide for future researchers if it emphasizes difference and origins of self-reproducible (or self-recoverable) concepts compared to other ML materials.

Response #6: We sincerely appreciate your kind suggestion on the terminology of ML. We fully agree that inconsistencies in terminology may cause confusion or misunderstanding for readers and researchers. Consequently, we have adopted the term "self-recoverable" instead of "self-reproducible" in the revised manuscript. After the revision, the terminology of ML is consistent with most of published literatures.

Meanwhile, based on your suggestion, we have emphasized the difference between the self-recoverable ML materials studied in this work and other types of ML materials. We believe that the additional explanation as you suggested is beneficial to avoid misunderstandings on different types of ML mechanisms. Please also see below cited from the manuscript.

“Here, it is worth emphasizing that the interfacial triboelectricity makes a clear contribution to the self-recoverable ML mechanism without the need of pre-excitation (generally using the triboelectricity induced electroluminescence model), but the effects of interfacial triboelectricity on other types of ML mechanisms (such as elasticoluminescence or fractoluminescence) are not discussed in this work. We believe that for different types of ML mechanisms, the dominant factors determining the ML intensity may vary greatly.” (Discussion part, Page 7)

Referee #2:

In this contribution, the authors establish a correlation between differences in triboelectric charge density and mechanoluminescence intensity in a series of inorganic/organic composite films built from apatite phosphors (doped with europium) and polymers (polydimethylsiloxane or silicone). This implies measuring the charges at the interface between the inorganic and organic parts, a task the authors achieved by the method used for preparing triboelectric nanogenerators described in the literature. Interestingly the mechanoluminescence is self-reproducible for all phosphors, so that the described systems have potentially interesting applications. For instance, color change that is demonstrated following self-redox processes affecting the europium valence. Electron transfer in multi-interface systems is also demonstrated. Finally, the presented data contribute to a better understanding of the mechanoluminescence generation mechanism in inorganic/organic composite materials since the presented data implying that it is most probably triboelectricity-induced electroluminescence.

Altogether, the work is well conducted, with control experiments, and the conclusions are backed by the reported data. It represents a novel and substantial contribution towards the design of highly luminescent self-reproducible mechanoluminescent materials so that I recommend publication.

Response: Thank you very much for your review and constructive comments on this work. We have carefully read your review and made corresponding revisions according to your kind comments and suggestions (please see below). We hope that after these revisions, the manuscript could be further improved.

Comment #1: Generally speaking, experimental uncertainties should be given on reported parameters (e.g. TECD, relative values S and ΔS , tensile intensity – error bars should be associated with data points, Fig. 2d,f - qPM)

Response #1: We appreciate your valuable suggestion on the experimental uncertainties of reported data. According to your suggestion, we have revised Fig. 2d-2f and Fig. 4c, 4d by adding error bars to the associated data points. The error bars represent the standard deviations (SD) of the results obtained from repeated tests. The revised figures are cited from the manuscript and shown below.

Fig. 2 | ML properties and relative triboelectric series difference (ΔS) in inorganics@organics ML materials containing apatite phosphors. (d) ML intensity of $MPX@PDMS$ under different tensile strains. (e) ML intensity of the inorganics@organics composites (I_{ML}), triboelectric transfer charge (q_{P-M}), relative triboelectric series difference of the ceramic/organic pairs (ΔS), and ML image during the first stretching. MPX [$M = C$ (Ca), S (Sr), B (Ba); $X = C$ (Cl), B (Br)] are abbreviations for $M_{5-x}(PO_4)_3X:Eu_x$ phosphors and ZnS is short for ZnS:Mn. Photographs of the inorganics@organics films being stretched are inserted in the figure. (f) Correlation between ΔS and ML intensity in the tested samples. The data presented in (d)-(f) are shown as the mean \pm 2 SD.

Fig. 4 | ML and interfacial electron transfer in multi-interface systems. (c) Triboelectric charge between the acrylic and different inorganic phosphors@PDMS composites. The triboelectric charge between acrylic and pure PDMS (q_{F-M0}) was also measured as a reference. MPX [$M = C$ (Ca), S (Sr), B (Ba); $X = C$ (Cl), B (Br)] are abbreviations for the $M_{5-x}(PO_4)_3X:Eu_x$ phosphors and ZnS is short for ZnS:Mn. (d) The correlation between the triboelectric transfer (between inorganic phosphors and organic matrix, q_{P-M}) and impact ML intensity. The ML intensity was recorded by impacting the inorganics@organics films with a falling acrylic ball. The size of the acrylic ball and the height of the ball above the films were fixed in the ML measurements. The data presented in (c) and (d) are shown as the mean \pm 2 SD.

Comment #2: Captions to Figs 1f,g and 2e. Please repeat abbreviations for ceramics MPX ($M = \text{C, Ca; S, Sr; B, Ba. X} = \text{B, Br; C, Cl}$) so that the captions are self-contained.

Response #2: Thank you very much for your kind suggestion on the sample abbreviations in figure captions. We understand that self-contained figure captions are important for readers to quickly get information from figures. We have repeated the abbreviations of MPX [$M = \text{C (Ca), S (Sr), B (Ba); X} = \text{B (Br), C (Cl)}$] for $M_{5-x}(\text{PO}_4)_3\text{X:Eu}_x$ in the captions of Fig. 1f, 1g, Fig. 2e, and Fig. 4c, 4d. The readers can now get the sample compositions from the captions without having to referring to other sections of the manuscript.

Comment #3: Page 4, The text starting with “The X-ray diffraction...”, left column and ending with “...mechanical stimulus (Figs 2e,2f).”, right column is entirely duplicated (end of page 4, page 5, beginning of page 6).

Response #3: Thank you very much for pointing out this problem. We apologize for causing this error due to our carelessness in preparing the manuscript. We have removed the duplicated text and carefully reviewed the entire manuscript to exclude other textual and formatting errors.

Comment #4: Experimental part: was the Eu content of the phosphors determined experimentally too?

Response #4: We are grateful for your concern on the experimental details. Basically, the Eu content of the phosphors (x ranging from 0.2 to 2.8) was calculated based on the stoichiometric ratio of raw materials. The actual content of Eu in the solid-state sintered phosphors is expected to be close to the nominal composition, given that no impurity phase was detected in the samples (Supplementary Fig. S17). Therefore, we did not measure the actual content of Eu in experiments, e.g. using the EDX method. We hope to get your understanding on this issue.

On the other hand, the Eu content of the phosphors used in the study of ML-triboelectricity correlation was indeed experimentally determined. For each combination of M and X elements in $M_{5-x}(\text{PO}_4)_3\text{X:Eu}_x$ ($M = \text{Ca, Sr, Ba; X} = \text{Br, Cl}$), one sample with a certain Eu content was chosen from the prepared phosphors based on the principle of the optimal PL intensity. This principle is adopted because it includes the effects of different host and different dopant content on the emission efficiency. We apologize for the unclear description in the previous experimental part, and we have made supplementary explanations in the revised manuscript. Please also see below the revision cited from the manuscript.

“For each combination of M and X elements in the $M_{5-x}(\text{PO}_4)_3\text{X:Eu}_x$, one sample with a certain Eu concentration was chosen from the obtained phosphors based on the principle of the optimal PL intensity (Supplementary Fig. S18). This principle is adopted because it includes the effects of different host and different dopant concentration on the emission efficiency. The Eu concentration (x value) for CPC, CPB, SPB, SCPC, BPB, SPC, SBPC, BPC used in the study of ML-triboelectricity correlation is 0.02, 0.04, 0.04, 0.03, 0.10, 0.06, 0.08, 0.10, respectively.” (Methods part, Page 9)

Referee #3:

The authors performed a systematic study on the interaction between inorganic phosphors and organic materials, in view of providing explanations for the mechanoluminescence behaviour upon application of stress. The emission mechanism of mechanoluminescence is still not very clear, especially regarding the “non-trap-based” phosphors that do not need previous excitation, before ML emission can occur. The reported experiments provide new elements for this explanation, although the exact mechanism is still not fully revealed (i.e. how the energy is transferred to the ML phosphors). Some balanced hypotheses are put forward. In my opinion, this is an important and smart work in the field and especially the proposed quantification method will have an impact in the research on mechanoluminescence.

In general, the manuscript is well written, although some grammatical errors are present. At some places, the wording can be improved. For instance, in the abstract “ambiguous mechano-to-photon conversion” is not really correct, as the conversion from stress to light emission is “poorly understood” or “not yet explained”, but not “ambiguous”. Similarly, in the second paragraph, this is called “fantastic mechano-to-photon conversion”. A more neutral tone would be better (also, I am not convinced already ‘thousands of phosphors’ have been shown to possess mechanoluminescence behaviour). Some terms are a bit strange, like “interface scale-up”, where essentially a large interface area is intended. Finally, regarding the structure of the paper, the section “ML properties in inorganics@organics composites” contains the same text twice.

Response: First of all, we sincerely appreciate your careful review on this work. Your suggestions and comments are very much helpful in improving our work. We have revised our manuscript and Supporting Information accordingly, and hope these revisions are appropriate also from your viewpoint.

We notice that there are many grammatical errors in the original manuscript. We apologize for these errors due to our carelessness in preparing the manuscript. We have corrected the errors you mentioned (please see below) and have carefully checked the entire manuscript. Thank you very much for your kind suggestion on the language.

“ambiguous mechano-to-photon conversion mechanism” is revised to “poorly understood mechano-to-photon conversion mechanism” (Abstract, Page 1)

“thousands of materials possessing fantastic mechano-to-photon conversion characteristics” is revised to “a large number of materials possessing mechano-to-photon conversion characteristics” (Paragraph 2, Page 1)

“interface scale-up” is revised to “interface amplification” (Figure 1, caption and the text)

The duplicate text in the section of “ML properties in inorganics@organics composites” is deleted.

Comment #1: I am confused about the consistency of some data sets in Figure 2. For instance, in part e, the BPC@PDMS is about 2.5 times stronger than the CPC@PDMS, whereas this is not the case in the data shown in part d. In part f, I would expect the values of the green bars in part e to be expressed as function of the values of the yellow bars in part e. However, if I look up the data points, it seems the blue bars are plotted (i.e. the triboelectric transfer charge, instead of the relative triboelectric series difference).

Response #1: Thank you very much for your comments on the data sets in Figure 2. In fact, the results in Fig. 2d and Fig. 2e were obtained in two independent experiments. In Fig. 2e, in order to quantitatively compare the ML intensity of different samples, the relative position of the sample and photodetector (and also the applied tensile strain) were strictly fixed for each test (sample). In Fig. 2d, the linear relationship between ML intensity and tensile strain of three samples was tested. However, the relative position of the sample and detector was not fixed in three sets of tests, and therefore the ML intensity of the three samples could not be compared. In this regard, we should not include the three sets of data into one figure,

because this may lead to misunderstanding of intensity comparison of the three samples. In response to this problem, we have divided Fig. 2d into three parts and provided coordinate scales for each one. Each part in the revised figure shows the linear relationship for one sample independently.

In Fig. 2f, we plotted the tensile ML intensity (the green bars & green label in Fig. 2e) as function of the relative triboelectric series difference (the blue bars with data in Fig. 2e). However, we mistakenly used a yellow label to represent the relative triboelectric series difference in the symbol description (the upper left corner of Fig. 2e). We have corrected this error and now the relative triboelectric series difference is shown in blue color (both in the symbol description and data). By the way, error bars are added for all the data in Fig. 2d-2f, as suggested by Reviewer 2.

Please refer to the revised Fig. 2d-2f as shown below. We sincerely apologize for the confusion generated due to our carelessness.

Fig. 2 | ML properties and relative triboelectric series difference (ΔS) in inorganics@organics ML materials containing apatite phosphors. (d) ML intensity of $MPX@PDMS$ under different tensile strains. (e) ML intensity of the inorganics@organics composites (I_{ML}), triboelectric transfer charge (q_{P-M}), relative triboelectric series difference of the ceramic/organic pairs (ΔS), and ML image during the first stretching. MPX [$M = C$ (Ca), S (Sr), B (Ba); $X = C$ (Cl), B (Br)] are abbreviations for $M_{5-x}(PO_4)_3X:Eu_x$ phosphors and ZnS is short for ZnS:Mn. Photographs of the inorganics@organics films being stretched are inserted in the figure. (f) Correlation between ΔS and ML intensity in the tested samples. The data presented in (d)-(f) are shown as the mean \pm 2 SD.

Comment #2: The manuscript mentions that the ML emission spectra are similar to the CL emission spectra, but both spectra (for each composition) are quite different, both in terms of the width and position of the Eu^{2+} emission band and of the presence or not of the Eu^{3+} emission. Especially the change in Eu^{2+} peak position is surprising. This point should be addressed in more detail. Also, adding (perhaps in the SI) for each phosphor composition a graph of PL, CL and ML emission would be very helpful, in order to easily assess the differences.

Response #2: We are really grateful for your constructive comments on the PL, CL, and ML spectra. Here, we need to admit that we had some errors in labelling the spectra in the previous Fig. 2a (ML) and Fig. 3c (CL), which led to the abnormal results in the comparison of ML and CL spectra. To resolve this problem, we re-measured all of the PL, CL, and ML spectra and re-plotted them in Fig. 2a (ML), Fig. 2c (CL) in a more accessible way. Meanwhile, we add a new figure to compare the three types of spectra for each phosphor composition according to your kind suggestion (Supplementary Fig. S7). We believe that the revision is much better for comparing these spectra. Thank you very much for your suggestion.

From Fig. 2a, 2c, and Supplementary Fig. S7, we can find the similarity between the ML and CL spectra, but they also show differences, especially in CSPC and SBPC. According to the PL spectra, Eu^{3+} and Eu^{2+} ions coexist in the phosphors containing Ca element, and only Eu^{2+} is present in other phosphors. Also, there are multiple bands of Eu^{2+} emissions in CSPS and SBPC, which could be attributed to multiple crystalline sites for Eu^{2+} substitution. We consider that the differences between the ML and CL spectra could be mainly due to the selective excitation of different emitting centers (Eu^{3+} and Eu^{2+} in different sites) under specific excitation conditions. In addition, the broadening of emission bands in the ML spectra could be also related to the use of a CCD-type fiber spectrometer with a large slit. Still, we know that completely elucidating the detailed differences of the spectra and clearly explaining the reasons requires further research and more efforts. In this work, if you agree, we would like to include the comparison results of the PL, CL, and ML spectra in Supporting Information (Supplementary Fig. S7) and briefly introduce them in the manuscript (cited from the manuscript and shown below).

“The photoluminescence (PL) spectra of the five phosphors MPX ($M = \text{Ca/Sr/Ba}$, $X = \text{Cl}$) confirm that Eu^{3+} may coexist and it tends to be present in the phosphors containing Ca (Supplementary Fig. S7). This is similar to the results of the ML spectra in Fig 2a. Meanwhile, multiple broad bands of Eu^{2+} emission are observed in the apatite phosphors, especially in CSPC and SBPC with two alkaline earth elements. Due to the coexistence of Eu^{3+} and Eu^{2+} (also including multiple crystalline sites for Eu^{2+}), the PL spectra show dependence on the excitation wavelength.” (Paragraph 6, Page 4)

“We further investigated the cathodoluminescence (CL) spectra of the phosphors under high-field electron bombardment (Fig. 2c and Supplementary Fig. S9). The CL spectra are basically consistent with the ML spectra (with some differences possibly due to selective excitation of different emitting centers), and the CL intensity increases with increasing the kinetic energy of the applied high-field electrons.” (Paragraph 7, Page 4)

Fig. 2 | ML properties and relative triboelectric series difference (ΔS) in inorganics@organics ML materials containing apatite phosphors. (a-c) ML spectra, TL glow curves and CL spectra of MPX@PDMS composites ($M = \text{Ca/Sr/Ba}$, $X = \text{Cl}$, luminescent center = Eu). The ML spectra were measured when 80% tensile strain was loaded on the MPX@PDMS films. The TL glow curves were recorded from 150 to 500 K with a fixed heating rate of 50 K/min. Prior to TL measurements, the samples were excited by 365-nm light for 20 s. The CL spectra were obtained by using scanning electron microscopy (SEM) with an electron beam of 7 kV and 60 mA.

Figure S7. PL, CL and ML spectra of MPX phosphors ($M = \text{Ca, Sr, Ba}$; $X = \text{Cl}$). (a-e) Comparison of emission spectra in each phosphor. The PL spectra of the samples were recorded under excitation of different wavelengths (labelled in the upper right corner). The PL spectra were measured with a multifunctional fluorescence spectrometer (FLS980, Edinburgh Instruments). The CL spectra were recorded by using a modified Mp-Micro-S instrument attached to the SEM (MonoCL4, Gatan). The ML spectra were recorded by using a fiber spectrometer covering the wavelength ranges of 300-1100 (QE Pro, Ocean Optics) under mechanical actions (e.g., pressing-releasing cycles). It should be noted that the use of the fiber spectrometer (CCD type) with a much larger slit results in the broadening of emission bands in the ML spectra.

Comment #3: The similarity between ML and CL spectra is used to speculate that the excitation is electrically induced. First, why is then the shape so different (i.e. in terms of the presence/absence of Eu^{3+} ; please show the CL spectra also up to 700 nm), and second, what is then the possible mechanism? Can it potentially be electron impact, due to accelerated charges, similar to (high field) electroluminescence, or is it rather e-h recombination, with the energy transferred to the luminescent ions?

Response #3: Thank you very much for your comments about the mechanism of electrically induced luminescence. According to your kind suggestion, we have re-measured the CL spectra up to 800 nm and plotted them in Fig. 2c. Furthermore, we investigated the CL spectra under different accelerating voltage of the applied high-field electron beam (Supplementary Fig. S9).

As described in the *Response of Comment #3*, the CL spectra are basically consistent with the ML spectra but with some differences due to the selective excitation of different emitting centers. Meanwhile, the CL intensity increases with the increase of the energy of the high-field electrons (Supplementary Fig. S9). These results support the impact excitation model of electrically induced luminescence in solids (J. Appl. Phys. 1991, 70: 4505), in which more hot electrons are generated when the impact energy increases and they further excite more luminescent centers. However, we understand that our work does not provide new insights about how the triboelectricity produces ML, and what we can do is to refer to the classic mechanism model of electroluminescence reported in literatures (i.e., the impact excitation model as you mentioned), and apply characterization methods to verify the applicability of this model to our study (e.g., Phys. Rev. B 2004, 69: 235109). We are happy to add some discussion based on our results. Please see the following discussion cited from the revised manuscript.

“Regarding the mechanism of electroluminescence under a high field, the impact excitation model has been considered to be one of the reasonable theories to explain this process. Our results show a positive correlation between the high-field electron energy and the CL intensity, which support the impact excitation model that more hot electrons are generated when the impact energy increases and they further excite more luminescent centers. Also, one can find that the ML intensity of the MPX@PDMS composites shows a nearly linear enhancement with increasing the tensile strain (Fig. 2d).” (Paragraph 7, Page 4)

Figure S9. Cathodoluminescence (CL) spectra of the MPX ($M = \text{Ca}, \text{Sr}, \text{Ba}; X = \text{Cl}$) phosphors by using different accelerating voltages of the electron beam. With the increase of the accelerating voltage from 3 to 7 kV, the CL intensity is increased correspondingly.

Comment #4: It would be very interesting to add the behaviour of SrAl₂O₄:Eu to the entire discussion (perhaps with the data only in the SI to keep the story consistent). Where is this phosphor positioned in the relative triboelectric series difference? When does it transition from trap-based ML to the “self-reproducible” ML?

Response #4: Thank you very much for your suggestion. This is indeed a very interesting discussion since SrAl₂O₄:Eu is the best known trap-based ML phosphor and self-recoverable ML in SrAl₂O₄:Eu has hardly been studied. We measured the triboelectric charge density (TECD) between a SrAl₂O₄:Eu ceramic sample and PDMS film by using the measurement system in Fig. 1b. The results in Supplementary Fig. S11 show that the maximum triboelectric charge and voltage between SrAl₂O₄:Eu and PDMS in the pressing-releasing cycles are much less than those between BPC (or ZnS) and PDMS. The calculated relative triboelectric series *S* value for SrAl₂O₄ is 1.81, smaller than 13.78 for BPC and 36.23 for ZnS. This indicates that the interfacial triboelectricity may also contribute to the generation of self-reproducible ML in SrAl₂O₄:Eu@PDMS, but its contribution should be much smaller than that in BPC@PDMS and ZnS@PDMS. The result also suggests that only when there are very few trapped charge carriers in SrAl₂O₄:Eu@PDMS (e.g., after heat bleaching at sufficiently high temperatures), self-recoverable ML might dominate over trap-based ML. However, this situation should be difficult to observe because of (i) the weak triboelectric effect in SrAl₂O₄:Eu@PDMS, (2) the large number of traps and highly efficient trap-based ML in SrAl₂O₄:Eu, and (3) the inability of PDMS to withstand high temperatures for thermal bleaching.

We have added the above results on the relative triboelectric series of SrAl₂O₄:Eu in the Supporting Information (Supplementary Fig. S10) and the manuscript according to your kind suggestion.

“In contrast, the calculated triboelectric series of SrAl₂O₄ relative to PDMS is only 1.81 (Supplementary Fig. S10). Considering the large number of traps and highly efficient trap-controlled ML in SrAl₂O₄:Eu phosphors, it would be difficult to observe dominant self-recoverable ML in SrAl₂O₄:Eu@PDMS.” (Paragraph 1, Page 5)

Figure S10. TECD test of a SrAl₂O₄:Eu ceramic plate and PDMS film by using the measurement system given in Fig. 1b. (a) Variations in triboelectric charge and (b) changes in voltage between the two materials in pressing-releasing cycles. The repetition frequency of the pressing-releasing cycles was 2.5 Hz. The calculated relative triboelectric series *S* value for SrAl₂O₄ is 1.81 according to Eq. (1) and (2).

Comment #5: The observation of the valence state changes in the ML upon repeated stretching is a very exciting result. It would be interesting to know if it is a permanent or temporary change, when looking at the CL or PL after e.g. 1000 stretches. Regarding the statement “to the best of our knowledge, the self-oxidation and self-reduction of Eu is firstly observed...”, I would guess a similar effect was already observed

in *Adv. Funct. Mater.* 2018, 28, 1803168.

Response #5: Thank you very much for your concern on the valence state changes of Eu. According to our experimental results, the PL spectra of the BPC@PDMS sample (after 2000 stretches) did not return to the original state after being kept at room temperature for 30 days (Supplementary Fig. 14). This indicates that the valence state changes are relatively stable (if not permanent) at the ambient condition. Nevertheless, we cannot rule out that the valence state changes may recover under high-temperature heat treatment. Unfortunately, we could not complete the high-temperature heat treatment experiment for verification, because PDMS could not withstand temperatures above 500 K.

Regarding the statement of self-oxidation and self-reduction of Eu, we have carefully read the paper you mentioned (*Adv. Funct. Mater.* 2018, 28: 1803168). Indeed, this paper shows interesting self-reduction of Eu in $\text{Sr}_3\text{Al}_2\text{O}_6:\text{Eu}^{3+}$ phosphors. However, the self-reduction of Eu is attributed to nonequivalent substitution and synthetic atmosphere control (cited from this paper: “It is interesting to observe that Eu^{3+} ions could spontaneously reduce to Eu^{2+} under an atmosphere of nitrogen (an inert environment with no oxidation and no reduction), which is called self-reduction. Such a phenomenon is believed to be aroused by interior charge self-regulation owing to the nonequivalent substitutions of Eu^{3+} to Sr^{2+} ”), rather than being generated when subjected to mechanical stimulus. In fact, this paper reports that the ML spectra in $\text{Sr}_3\text{Al}_2\text{O}_6:\text{Eu}^{3+}$ remains almost unchanged after 1000 cycles of stretching (see the figure cited from this paper and reproduced here as Figure R1). Therefore, the results reported in this paper is markedly different from the phenomenon of self-oxidization and self-reduction of Eu during mechanical stimulus found in our work. Nevertheless, we feel that the statement in the original manuscript may cause misunderstanding and therefore we would like to modify the statement in the revised manuscript. Please see the revision cited from the manuscript.

“To the best of our knowledge, the self-oxidization and self-reduction of Eu upon continuous mechanical stimulus is firstly observed in ML materials. This interesting phenomenon may also provide solid evidence for the electron transfer process in ML inorganics@organics composites.” (Paragraph 2, Page 6)

Figure S13. PL spectra of the BPC@PDMS sample at different stages. (a) As-prepared BPC@PDMS composite film without stretching. (b) BPC@PDMS after 2000 stretches. (c) BPC@PDMS after 2000 stretches and then kept at room temperature for 30 days. The excitation wavelength is 322 nm.

Figure R1. Initial ML spectra of $\text{Sr}_3\text{Al}_2\text{O}_6:\text{Eu}^{3+}@\text{PDMS}$ and the ML spectra of the sample after 1000 cycles of stretching. This figure is cited from *Adv. Funct. Mater.* 2018, 28: 1803168.

Comment #6: Regarding the self-oxidation and self-reduction, it would be interesting to add what happens to the phosphors closer to the zero value for S , but with reasonable ML intensity (e.g. for SPC, or even CPB/SPB). Is the valence state stable in that case?

Response #6: Thank you very much for the interesting question. According to your comments, we further measured the ML change of SPC@PDMS under repeated stretching-releasing cycles. The relative triboelectric series of SPC is ~ 1.18 , which is close to the zero point of PDMS (Fig. 1g). As shown in Supplementary Fig. S11, S12, after the sample was continuously stretched, additional Eu^{3+} peaks appear in the ML spectra, accompanied by an overall decrease in intensity. This verifies that the oxidation of Eu ions also occurs in SPC@PDMS, but the degree of valence state change is less than that in BPC@PDMS. The ML spectra recorded after the stretched sample was further kept at the ambient condition for 1 day (Supplementary Fig. S12c), which indicates that the valence state change is stable at room temperature. This is similar to the case of BPC@PDMS. We have added the above results on SPC@PDMS into the Supporting Information as Fig. S11, S12. Thank you for your suggestion.

Figure S11. Photographs of the SPC@PDMS film under stretching for 2000 times. The ML was partially converted from blue (Eu^{2+}) to red (Eu^{3+}) under the mechanical stimulus.

Figure S12. ML spectra of the SPC@PDMS film (a) under the first stretch, (b) after ~2000 stretches, and (c) further kept at ambient condition for 16 h. The insets show the enlarged part of the spectra focusing on Eu^{3+} emissions. Comparison between (a) and (b) suggests that a small part of Eu^{2+} is oxidized to Eu^{3+} during the continuous stretching. Comparison between (b) and (c) indicates that the valence state change is stable at room temperature.

Comment #7: As far as I know, the abbreviation SC for silicone is not very common. So, it would be interesting to also mention this in Figure 1, or in its caption.

Response #7: Thank you again for your kind suggestion. We have added a note about this abbreviation in the caption of Figure 1. We believe this will be useful for readers. Thank you very much.

Comment #8: Estimating the transferred charges is not straightforward, especially for more complex structures. What is the motivation for the validity of equation 3, considering limited charge mobility in most of the constituent components of the multi-interface system in Fig. 4b?

Response #8: Thank you very much for your valuable comments on the transferred charge. We fully agree with you that the charge mobility is quite small in the studied system. In this regard, Equation 3 is a simple model that describes the influence of the triboelectricity at the P-M interface ($q_{\text{P-M}}$) on the transferred charges at the F-M interfaces (i.e., the difference between $q_{\text{F-M0}}$ and $q_{\text{F-M}}$, or $q_{\text{F-M}} - q_{\text{F-M0}}$). We understand that not all of the triboelectricity at the P-M interface ($q_{\text{P-M}}$) could contribute to the F-M interfaces due to the limited charge mobility, which may result in an underestimation of their contribution. Here, we introduce a factor k to describe the incomplete contribution of the triboelectricity at the P-M interfaces as shown in the revised Equation 3. Considering the same PDMS matrix and phosphor/PDMS ratio, we assume that the k factor is a constant in different composite materials and thus the triboelectric charge between the acrylic and MPX@PDMS under mechanical stimulus ($q_{\text{F-M}}$) can be estimated by using Equation 3.

We believe that Equation 3 could be helpful to describe the contribution of the triboelectricity in multi-interface systems. But we also know that Equation 3 is an over-simplified model and it undoubtedly needs further study in the future. We hope to get your kind understanding on our difficulty in clearly addressing this issue presently. The revision on Equation 3 is cited from the manuscript and shown below.

“The total amount of transferred charges is expressed by:

$$q_{F-M} = q_{F-M0} + k \cdot q_{P-M} \quad (3)$$

Here, k is a factor to describe the incomplete contribution of the triboelectricity at the P-M interfaces due to the limited charge mobility in the constituent components. We assume that the k factor is a constant in different composite materials (with the same PDMS matrix and the same phosphor/PDMS ratio), the triboelectric charge between the acrylic and MPX@PDMS under mechanical stimulus (q_{F-M}) can be estimated and the results are present in Fig. 4c and Supplementary Fig. S14.” (Paragraph 1, Page 7)

Comment #9: The experimental section mentions calculations of the Fermi level. However, unless I missed it, I didn't see results from those calculations.

Response #9: Thank you very much for pointing out this problem. In fact, we had tried to calculate the Fermi levels of the interfacial structures between the phosphors and PDMS. The results seemed interesting, but finally we failed to establish a clear relationship between the Fermi levels and the recorded triboelectric transfer charge. Therefore, we decided not to include the calculations of the Fermi levels in this work and remove these results from the manuscript. However, we carelessly left the introduction of the Fermi level calculations in the experimental section. We have deleted this part from the revised manuscript.

Comment #10: The data shown in the SI is relevant. There is a good balance between the data shown in the main manuscript and in the supporting information.

Response #10: We appreciate your positive comments on this work. During this revision, we have added new data to the Supporting Information and the main manuscript as suggested by you and other reviewers. We hope this does not break the balance of the data.

Once again, thank you very much for your constructive comments and suggestions. Your comments and suggestions are valuable for us to further improve this work.

Reviewer #1 (Remarks to the Author):

The authors have addressed the reviewer's comments and concerns and have improved the quality of the paper accordingly. Therefore, I recommend the publication of this manuscript in Nature Communications.

Reviewer #2 (Remarks to the Author):

A careful reading of the revised manuscript and of the explanations given by the authors show that remarks from all reviewers have been taken seriously into consideration. The text has been modified and, most importantly, several new data have been added. The work is now much better, compared to the initial submission, it describes important findings that will be most helpful for further developments of self-recoverable ML; therefore in my opinion it deserves publication.

Remark

I have only one technical remark in that I do not understand Figure S11 in that I do not really see red ML (apart for a tendency to shift from crude blue to blue-violet)? Maybe could the authors think of another way of presenting it, so that the red luminescence would be better evidenced.

Reviewer #3 (Remarks to the Author):

The authors have responded in great detail to the reviewer comments. Additional experiments have been performed. The results have been integrated in the manuscript and in the SI. Some inconsistencies and data processing errors have been fixed. In my opinion, the manuscript can now be accepted for publication. I am still convinced the publication will have an impact in the field.

I just wanted to briefly comment on Comment 5 (reviewer 3). From the information in the cited source it is indeed correct that there is no information on a permanent change in valence state (as monitored via the PL before and after, for instance). Only during the dynamic stretching, the $\text{Eu}^{2+}/\text{Eu}^{3+}$ ratio changes, depending on the strain. I want to mention explicitly that I agree with the authors of the current manuscript that this is indeed not the same (although both effects might be coupled, if a less stable valence change occurs) and that I should not have compared the two situations.

Point-by-Point Response to Referees

Referee #1:

The authors have addressed the reviewer's comments and concerns and have improved the quality of the paper accordingly. Therefore, I recommend the publication of this manuscript in *Nature Communications*.

Response: Thank you very much for your positive evaluation and the recommendation for publication. We are grateful to you for giving all the valuable suggestions and comments for this work.

Referee #2:

A careful reading of the revised manuscript and of the explanations given by the authors show that remarks from all reviewers have been taken seriously into consideration. The text has been modified and, most importantly, several new data have been added. The work is now much better, compared to the initial submission, it describes important findings that will be most helpful for further developments of self-recoverable ML; therefore in my opinion it deserves publication.

I have only one technical remark in that I do not understand Figure S11 in that I do not really see red ML (apart for a tendency to shift from crude blue to blue-violet)? Maybe could the authors think of another way of presenting it, so that the red luminescence would be better evidenced.

Response: Thank you very much for your positive comments on our revision. We are pleased that our revision is accepted to you. Thank you again for your constructive and kind suggestions during the whole review process.

Regarding your concern about Supplementary Figure 11, we fully agree with you that red ML could be hardly observed from the photographs. The emission of Eu^{3+} , even in the case after 2000 stretches, was submerged by the much more intense emission of Eu^{2+} . This is understandable because the triboelectric series of SPC (1.18) is close to that of PDMS (see Fig. 1g). Still, self-oxidization of Eu^{2+} indeed occurred in SPC@PDMS upon repeated stretching, which is evidenced in the ML spectra in Supplementary Figure 12b (indicating Eu^{3+} emission). We believe that the best way to present Eu^{3+} emission is showing the ML spectra. Therefore, we would like to revise the figure caption of Supplementary Figure 11 (not to mention red ML) and describe the ML variation in combination with the results of the ML spectra (Figure 12b).

Referee #3:

The authors have responded in great detail to the reviewer comments. Additional experiments have been performed. The results have been integrated in the manuscript and in the SI. Some inconsistencies and data processing errors have been fixed. In my opinion, the manuscript can now be accepted for publication. I am still convinced the publication will have an impact in the field.

I just wanted to briefly comment on Comment 5 (reviewer 3). From the information in the cited source it is indeed correct that there is no information on a permanent change in valence state (as monitored via the PL before and after, for instance). Only during the dynamic stretching, the $\text{Eu}^{2+}/\text{Eu}^{3+}$ ratio changes, depending on the strain. I want to mention explicitly that I agree with the authors of the current manuscript that this is indeed not the same (although both effects might be coupled, if a less stable valence change occurs) and that I should not have compared the two situations.

Response: We are grateful to you for your careful review and constructive feedback, which have greatly contributed to the improvement of the manuscript. We also appreciate your comment regarding the valence state change and your kind understanding on our explanation. Once again, we extend our deepest thanks to you for your constructive suggestion and guidance throughout this review process.